# Weakly-Supervised Concealed Object Segmentation with SAM-based Pseudo Labeling and Multi-scale Feature Grouping

**Chunming He**[1,*] , **Kai Li**[2,*] , **Yachao Zhang**[1] , **Guoxia Xu**[3] ,
**Longxiang Tang**[1] , **Yulun Zhang**[4] , **Zhenhua Guo**[5] , and **Xiu Li**[1,†]
[1]Shenzhen International Graduate School, Tsinghua University, [2]NEC Laboratories America,
[3]Nanjing University of Posts and Telecommunications, [4]ETH Zürich, [5]Tianyi Traffic Technology

## Abstract

Weakly-Supervised Concealed Object Segmentation (WSCOS) aims to segment objects well blended with surrounding environments using sparsely-annotated data for model training. It remains a challenging task since (1) it is hard to distinguish concealed objects from the background due to the intrinsic similarity and (2) the sparsely-annotated training data only provide weak supervision for model learning. In this paper, we propose a new WSCOS method to address these two challenges. To tackle the intrinsic similarity challenge, we design a multi-scale feature grouping module that first groups features at different granularities and then aggregates these grouping results. By grouping similar features together, it encourages segmentation coherence, helping obtain complete segmentation results for both single and multiple-object images. For the weak supervision challenge, we utilize the recently-proposed vision foundation model, "*Segment Anything Model (SAM)*", and use the provided sparse annotations as prompts to generate segmentation masks, which are used to train the model. To alleviate the impact of low-quality segmentation masks, we further propose a series of strategies, including multi-augmentation result ensemble, entropy-based pixel-level weighting, and entropy-based image-level selection. These strategies help provide more reliable supervision to train the segmentation model. We verify the effectiveness of our method on various WSCOS tasks, and experiments demonstrate that our method achieves state-of-the-art performance on these tasks. The code will be available at https://github.com/ChunmingHe/WS-SAM.

## 1 Introduction

Concealed object segmentation (COS) aims to segment objects visually blended with surrounding environments [1]. COS is a general term with different applications, e.g., camouflaged object detection [2, 3], polyp image segmentation [4, 5], transparent object detection [6, 7], etc. COS is a challenging task due to the intrinsic between foreground objects and background, which makes it extremely difficult to identify discriminative clues for accurate foreground-background separation. To address this challenge, existing methods have employed approaches that mimic human vision [8–10], introduce frequency information [11, 12], or adopt joint modeling in multiple tasks [13–18].

Weakly-Supervised COS (WSCOS) studies an even more challenging yet more practical problem, involving learning a COS model without relying on pixel-wise fully-annotated training data. WSCOS greatly reduces annotation costs by only requiring a few annotated points or scribbles in the foreground

---

*Equal Contribution, † Corresponding Author

37th Conference on Neural Information Processing Systems (NeurIPS 2023).

or background. However, the sparsity of annotated training data diminishes the limited discrimination capacity of the segmenter during model learning, thus further restricting segmentation performance.

In this paper, we propose a new algorithm for the challenging WSCOS task. To tackle the intrinsic similarity of foreground and background, we introduce a Multi-scale Feature Grouping (MFG) module that first evacuates discriminative cues at different granularities and then aggregates these cues to handle various concealing scenarios. By performing feature grouping, MFG essentially promotes coherence among features and thus is able to alleviate incomplete segmentation by encouraging local correlation within individual objects, while also facilitating multiple-object segmentation by seeking global coherence across multiple objects.

To address the challenge of weak supervision, we propose to leverage the recently proposed vision foundation model, *Segment Anything Model (SAM)*, to generate dense masks by using sparse annotations as prompts, and use the generated masks as pseudo labels to train a segmenter. However, due to the intrinsic similarity between foreground objects and the background, the pseudo labels generated by SAM may not always be reliable. We propose a series of strategies to address this problem. First, we propose to generate multiple augmentation views for each image and fuse the segmentation masks produced from all views. The fused mask can highlight reliable predictions resistant to image augmentations and tend to be more accurate and complete as an effect of ensemble. Second, we propose an entropy-based weighting mechanism that assigns higher weights to predictions of pixels of high certainty. Lastly, to deal with the extreme images that SAM fails to generate reasonably correct masks, we propose an entropy-based image-level selection technique to assess the quality of the generated mask and decide whether to use the masks as pseudo labels for model training. These strategies ensure that only high-quality pseudo labels are used for training the segmenter. For ease of description, we refer to our solution of using SAM to address this task as *WS-SAM*.

Our contributions are summarized as follows:

(1) We propose to leverage SAM for weakly-supervised segmentation by using the provided sparse annotations as prompts to generate dense segmentation masks and train a task segmentation model. To the best of our knowledge, this is the first attempt to leverage the vision foundation model to address the weakly-supervised segmentation task.

(2) We propose a series of strategies for dealing with potentially low segmentation mask quality, including the multi-augmentation result ensemble technique, entropy-based pixel-level weighting technique, and entropy-based image-level selection technique. These techniques help provide reliable guidance to train the model and lead to improved segmentation results.

(3) We introduce a Multi-scale Feature Grouping (MFG) technique to tackle the intrinsic similarity challenge in the WSCOS task. MFG evacuates discriminative cues by performing feature grouping at different granularities. It encourages segmentation coherence, facilitating to obtain complete segmentation results for both single and multiple object images.

(4) We evaluate our method on various WSCOS tasks, and the experiments demonstrate that our method achieves state-of-the-art performance.

## 2  Related Works

**Segment Anything Model**. SAM [19] is a recently proposed vision foundation model trained on SA-1B of over 1 billion masks. Its primary objective is to segment any object in any given image without requiring any additional task-specific adaptation. Its outstanding quality in segmentation results and zero-shot generalization to new scenes make SAM a promising candidate for various computer vision tasks. However, recent studies have highlighted that SAM encounters difficulties when segmenting objects with poor visibility, such as camouflaged objects [20–22], medical polyps [23–27], and transparent glasses [28, 22]. These findings suggest that SAM still has limitations in COS tasks.

In this paper, we propose using SAM to generate dense segmentation masks from sparse annotations and introduce the first SAM-based weakly-supervised framework in COS, termed WS-SAM. To further increase the accuracy of the generated pseudo-labels, we propose a pseudo label refinement strategy. Such a strategy assigns higher weights to those reliable predictions that are resistant to various image augmentation. Therefore, WS-SAM can offer more precise and stable guidance for the learning process, ultimately boosting the segmentation performance of the segmenter.

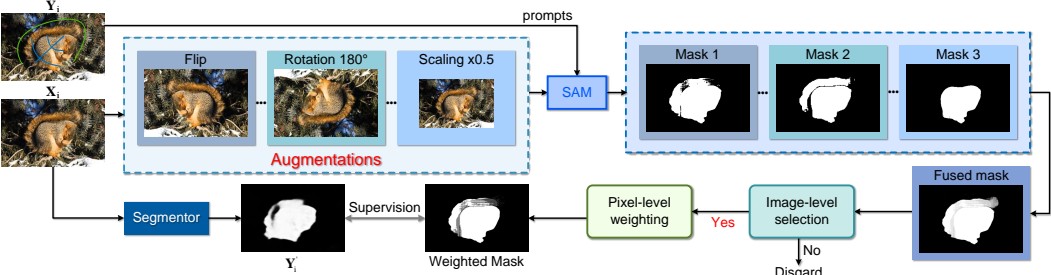

Figure 1: Framework of WS-SAM with scribble supervision. Note that the corresponding masks of the augmented images are inversely transformed so as to be consistent with the original image.

**Concealed Object Segmentation**. With the rapid development of deep learning, learning-based segmenters have obtained great achievements in the fully-supervised COS tasks [4, 2, 29]. PraNet [4] proposed a parallel reverse attention network to segment polyps in colonoscopy images. Drawn inspiration from biology, SINet [2] designed a predator network to discover and locate camouflaged objects. To detect transparent objects, GSDNet [29] integrated an aggregation module and a reflection refinement module. However, there is limited research on the weakly-supervised COS task. SCOD [30] introduced the first weakly-supervised COD framework, but it is only supervised with sparse annotations, which greatly restricts its discrimination capacity and inevitably inhibits segmentation performance. To address this problem, we first propose using SAM to generate precise pseudo-labels. Besides, to tackle intrinsic similarity, we introduce the multi-scale feature grouping module to evacuate discriminative cues at different granularities and thus promote feature coherence.

## 3  Methodology

Weakly-Supervised Concealed Object Segmentation (WSCOS) aims to learn a segmentation model from a sparsely-annotated training dataset $\mathcal{S} = \{\mathbf{X}_i, \mathbf{Y}_i\}_{i=1}^{S}$ and test the model on a test dataset $\mathcal{T} = \{\mathbf{T}_i\}_{i=1}^{T}$, where $\mathbf{X}_i$ and $\mathbf{T}_i$ denote the training and test images, respectively; $\mathbf{Y}_i$ represents the sparse annotations, which could be a few points or scribbles annotated as foreground or background.

Learning the segmentation model could be a challenging task, as concealed objects usually blend well with their surrounding environment, making it hard to distinguish foreground from background. Besides, the sparse annotations $\mathbf{Y}_i$ may not provide sufficient supervision to learn the model capable of making accurate dense predictions. To address these challenges, we first propose a strategy of leveraging the recently-proposed vision foundation model, *Segment Anything Model (SAM)*, to generate high-quality dense masks from sparse annotations and use the dense masks as pseudo labels to train the segmentation model. In addition, we propose a Multi-scale Feature Grouping (MFG) module that groups features at different granularities, encouraging segmentation coherence and facilitating obtain complete segmentation results for various concealing scenarios.

### 3.1  Pseudo Labeling with SAM

SAM is a recently-released vision foundation model for generic object segmentation [19]. It is trained with more than one billion segmentation masks and has shown impressive capabilities of producing precise segmentation masks for a wide range of object categories (so-called "*segment anything*"). Unlike some enthusiasts who brag SAM has "killed" the segment task, we find that SAM is far from reaching that level, at least for the studied concealed object segmentation task. This is first because SAM requires "prompts" that provide clues about the objects of interest to produce segmentation results. While the prompts could be in many forms, e.g., points, masks, bounding boxes, etc., they are required to be provided by humans or other external sources (e.g., other algorithms) [2]. This requirement of the additional prompt inputs makes SAM unable to be (directly) used for applications where only test images are provided. In addition, we find that while SAM exhibits

---

[2]SAM includes an automatic prompt generation mechanism that first densely selects points across the whole image as prompts, then generates results based on the dense prompts, and last fuse the results. The problem is it segments everything possible in the image, with no distinguishing of objects of interest and others.

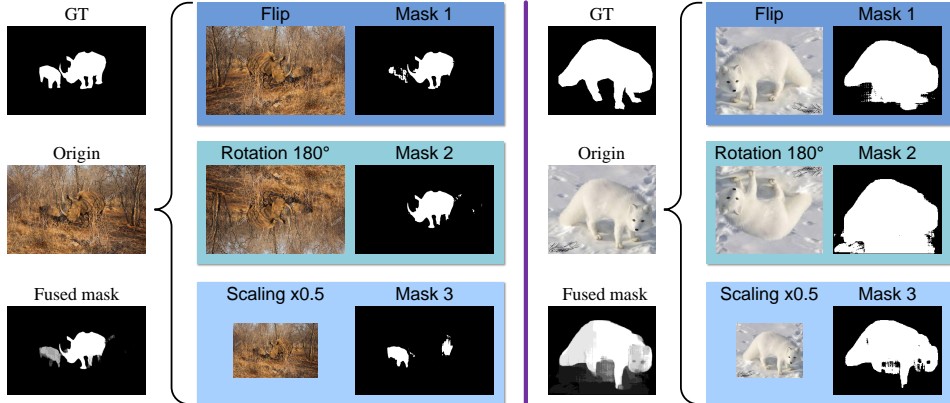

Figure 2: Masks of SAM with different augmented images. We inversely transform the masks to keep consistent with the original image. It is observed that fused masks contain more accurate and complete segmentation information.

impressive performance for general scene images, it still struggles for concealed object images, due to the intrinsic similarity between foreground objects and the background.

In this paper, we introduce SAM for the Weakly-Supervised Concealed Object Segmentation (WS-COS) task. As shown in Fig. 1, we use SAM to generate segmentation masks on training images by taking the sparse annotations as prompts, and take the segmentation masks as pseudo labels to train a COS model, which will be used for the test. It is expected that the SAM-generated pseudo labels are not reliable. We address this problem by proposing three techniques, namely, Multi-augmentation result fusion, and pixel-level weighting and image-level selection.

**Multi-augmentation result fusion**. Given a concealed image $(\mathbf{X}_i, \mathbf{Y}_i) \in \mathcal{S}$, we generate $K$ augmented images $\{\mathbf{X}_i^k\}_{k=1}^K$ by applying stochastic augmentations randomly sampled from image flipping, rotation $(0°, 90°, 180°, 270°)$, and scaling $(\times 0.5, \times 1.0, \times 2.0)$. We send $\{\mathbf{X}_i^k\}_{k=1}^K$ to SAM by using the sparse annotations $\mathbf{Y}_i$ as prompts, and generate segmentation masks $\{\mathbf{M}_i^k\}_{k=1}^K$, where

$$\mathbf{M}_i^k = \text{SAM}\left(\mathbf{X}_i^k, \mathbf{Y}_i\right). \tag{1}$$

Note that $\mathbf{M}_i^k$ has the same shape as input image $\mathbf{X}_i^k$, which may differ in shape from $\mathbf{X}_i$; we perform inverse image transformation to ensure all masks have the same shape as the original image.

As different segmentation results can be obtained when feeding SAM with different prompts, we expect $\{\mathbf{M}_i^k\}_{k=1}^K$ to vary since different augmented images are used for segmentation. Fig. 2 shows some examples. We can see that while these masks vary significantly in shape, they overlap in certain regions, which are reliably predicted by SAM regardless of image transformations and usually correspond to correctly predicted foreground regions. Besides, these masks complement each other, such that some foreground regions missed by one mask can be found in other masks. Based on these observations, we propose to fuse the segmentation masks for different augmented images, as

$$\tilde{\mathbf{M}}_i = \frac{1}{K} \sum_{k=1}^K \mathbf{M}_i^k, \tag{2}$$

where $\tilde{\mathbf{M}}_i$ is the fused mask. We expect $\tilde{\mathbf{M}}_i$ to be more reliable than the individual masks as it is an ensemble over various augmented images.

**Pixel-level weighting**. The prediction reliability of different pixels may vary. To highlight those more reliable ones, we propose to use entropy to weight the predictions. We calculate the entropy of each pixel and get an entropy map as

$$\tilde{\mathbf{E}}_i = -\tilde{\mathbf{M}}_i \log \tilde{\mathbf{M}}_i - (1 - \tilde{\mathbf{M}}_i) \log(1 - \tilde{\mathbf{M}}_i). \tag{3}$$

As the entropy map is calculated from the fused mask, it measures the prediction uncertainty of each pixel across all augmentation as a pixel will have low entropy only if it is *confidently and meanwhile consistently* predicted from all augmented images. Therefore, we can use this entropy map to weigh the fused mask $\mathbf{M}_i^k$ and assign higher weights to those reliable pixels.

**Image-level selection**. We have observed that for some highly challenging concealed images, SAM fails to produce even reasonably correct results with the sparse annotations as the prompts, with

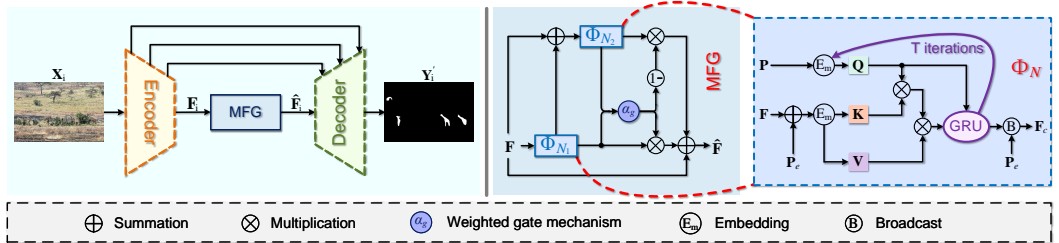

Figure 3: Architecture of the proposed model. $\Phi_P$ denotes feature grouping with $P$ prototypes. We simplify the broadcast process in $\Phi_P$ for space limitation.

whatever types of augmented images. This fundamentally invalidates the above pixel-wise weighting strategy. To deal with this case, we further propose an image-level selection mechanism to selectively choose images for training, further striving to provide reliable supervision to train the segmenter.

Similar to using entropy to define the pixel-level prediction uncertainty, we propose two entropy-based image-level uncertainty measurements, namely, absolute uncertainty $U_a$ and relative uncertainty $U_r$. Absolute uncertainty $U_a$ refers to the proportion of high-uncertainty pixels among all pixels, while relative uncertainty $U_r$ indicates the proportion of high-uncertainty pixels and foreground pixels with low uncertainty, which is specifically designed to accommodate small-object scenarios. We regard a pixel as a high-uncertainty pixel when its entropy is above 0.9. We define the following indicator function to decide whether to keep an image for training or not:

$$\hat{\mathbf{M}}_i = \mathbb{1}[U_a < \tau_a] \times \mathbb{1}[U_r < \tau_r], \tag{4}$$

where $\tau_a$ and $\tau_r$ are the thresholds that are set as 0.1 and 0.5 in this paper, respectively.

Applying the entropy weights $\tilde{\mathbf{E}}_i$ on the image selection indicator $\hat{\mathbf{M}}_i$, we reach our final mask that will be used for training the segmenter as,

$$\hat{\mathbf{Y}}_i = (1 - \tilde{\mathbf{E}}_i) \times \hat{\mathbf{M}}_i. \tag{5}$$

Our technique leverages SAM to generate segmentation masks and further incorporates multi-augmentation result fusion, pixel-level uncertainty weighting, and image-level uncertainty filtering, thus being able to generate reliable pseudo labels to train the segmenter.

## 3.2 Multi-scale Feature Grouping

The intrinsic similarity in concealed objects may cause incomplete segmentation and partial object localization in multi-object segmentation. Such problems could be further aggravated in weakly supervised scenarios due to the limited discriminative capacity of the segmenter. To address this issue, we propose a Multi-scale Feature Grouping (MFG) module that evacuates discriminative cues at various granularities. MFG achieves this by exploring the coherence of foreground/background regions and performing feature grouping at different levels. By encouraging feature coherence, MFG can alleviate incomplete segmentation by enhancing local correlation within individual objects and further facilitate multiple-object segmentation by seeking global coherence across multiple objects. The architecture of the proposed MFG module is illustrated in Fig. 3.

**Feature grouping**. Suppose $\mathbf{F} \in \mathbb{R}^{H \times W \times C}$ is the feature representation of an input image. We perform feature grouping by mapping $\mathbf{F}$ to $N$ learnable cluster prototypes $\mathbf{P} \in \mathbb{R}^{N \times C}$. These cluster prototypes $\mathbf{P}$ are randomly initialized. We first append the learnable spatial positional embedding $\mathbf{P}_e$ to the input feature $\mathbf{F}$ and get $\mathbf{F}_p$. Then, we linearly transform the prototypes $\mathbf{P}$ and the positioned feature $\mathbf{F}_p$ into $\mathbf{Q} \in \mathbb{R}^{N \times C}$, $\mathbf{K} \in \mathbb{R}^{HW \times C}$, and $\mathbf{V} \in \mathbb{R}^{HW \times C}$:

$$\mathbf{Q} = \mathbf{W}_q \mathbf{P}, \quad \mathbf{K} = \mathbf{W}_k \mathbf{F}_p, \quad \mathbf{V} = \mathbf{W}_v \mathbf{F}_p, \tag{6}$$

where $\mathbf{W}_q, \mathbf{W}_k, \mathbf{W}_v \in \mathbb{R}^{C \times C}$ are the learnable weights. To ensure the exclusive assignment of features to the cluster prototypes, we normalize the coefficients over all prototypes,

$$\bar{\mathbf{A}}_{i,j} = \frac{e^{\mathbf{A}_{i,j}}}{\sum_l e^{\mathbf{A}_{i,l}}}, \quad \text{where} \quad \mathbf{A} = \frac{1}{\sqrt{C}} \mathbf{K}^\top \mathbf{Q}. \tag{7}$$

We then calculate the integral value $\mathbf{U}$ of the input values with respect to the prototypes as

$$\mathbf{U} = \mathbf{D}^\top \mathbf{V}, \quad \text{where} \quad \mathbf{D}_{i,j} = \frac{\mathbf{A}_{i,j}}{\sum_l \mathbf{A}_{i,l}}, \tag{8}$$

and update the prototypes $\mathbf{P}$ by feeding $\mathbf{P}$ and $\mathbf{U}$ into a Gated Recurrent Units $GRU(\cdot)$:
$$\mathbf{P} = GRU\left(inputs = \mathbf{U}, states = \mathbf{P}\right). \tag{9}$$
By repeating Eqs. (6) - (9) for $T$ iterations, the cluster prototypes are iteratively updated and gradually strengthen the association between similar features, where $T = 3$ in this paper.

We broadcast each prototype onto a 2D grid augmented with the learnable spatial position embedding $\mathbf{P}_e$ to obtain $\{\mathbf{F}_i'\}_{i=1}^N \in \mathbb{R}^{H \times W \times C}$, and use $1 \times 1$ convolution to downsample each prototype, obtaining $\{\mathbf{F}_i''\}_{i=1}^N \in \mathbb{R}^{H \times W \times C/N}$. We concatenate those prototypes and obtain $\mathbf{F}_c \in \mathbb{R}^{H \times W \times C}$.

For ease of future use, we denote the feature grouping process with $N$ prototypes as $\mathbf{F}_c = \Phi_N(\mathbf{F})$.

**Multi-scale feature aggregation**. The number of prototypes $N$ in the above feature grouping technique controls the grouping granularity: a smaller value of $N$ facilitates the extraction of global information, while a larger value of $N$ can provide more valuable detailed information. To strike a balance, we propose to aggregate the multi-scale grouping features with different numbers of prototypes. Taking inspiration from the second-order Runge-Kutta (RK2) structure known for its superior numerical solutions compared to the traditional residual structure [31, 32], we employ RK2 to aggregate those features. Additionally, as shown in Fig. 3, we adopt a weighted gate mechanism $\alpha_g$ to adaptively estimate the trade-off parameter rather than using a fixed coefficient. Given the feature $\mathbf{F}$, the adaptively aggregated feature $\hat{\mathbf{F}}$ is formulated as follows:
$$\hat{\mathbf{F}} = \mathbf{F} + \alpha_g \Phi_{N_1}(\mathbf{F}) + (1 - \alpha_g)\Phi_{N_2}(\mathbf{F} + \Phi_{N_1}(\mathbf{F})), \tag{10}$$
where $\alpha_g = S(\sigma\, cat(\Phi_{N_1}(\mathbf{F}), \Phi_{N_2}(\mathbf{F} + \Phi_{N_1}(\mathbf{F}))) + \mu)$. $S$ is Sigmoid. $\sigma$ and $\mu$ are the learnable parameters in $\alpha_g$. $N_1$ and $N_2$ are the numbers of groups, which are empirically set as 4 and 2.

Our multi-scale feature grouping technique is inspired by the slot attention technique [33], but we differ from slot attention in the following aspects. Slot attention targets at instance-level grouping in a self-supervised manner, our MFG is proposed to adaptively excavate the feature-level coherence for complete segmentation and accurate multi-object localization. To relax the segmenter and ensure generalization, we remove the auxiliary decoder used in slot attention for image reconstruction, along with the reconstruction constraint. Additionally, we employ an RK2 structure to aggregate the multiscale grouping feature with different numbers of prototypes, which further facilitates the excavation of feature coherence and therefore helps improve segmentation performance.

## 3.3 Weakly-Supervised Concealed Object Segmentation

To use the proposed MFG technique for concealed object segmentation, we integrate MFG with the encoder and decoder architecture utilized in an existing camouflaged object detection model [12] to construct a novel segmenter. The model comprises a ResNet50-backed encoder $E$ that maps an input image $\mathbf{X}_i$ to a feature space as $\mathbf{F}_i = E(\mathbf{X}_i)$. Using the obtained $\mathbf{F}_i$, we apply MFG to perform multi-scale feature grouping, resulting in $\hat{\mathbf{F}}_i = MFG(\mathbf{F}_i)$. Subsequently, a decoder $D$ maps $\hat{\mathbf{F}}_i$ back to the image space, generating the predicted mask $\mathbf{Y}_i' = D(\hat{\mathbf{F}}_i)$. Fig. 3 provides a conceptual illustration of this model, and *more architecture details can be found in the supplementary materials.*

Following [30], we train the whole model jointly with the sparse annotations $\mathbf{Y}_i$, and the generated segmentation mask $\hat{\mathbf{Y}}_i$ and $\tilde{\mathbf{M}}_i$ by the SAM model, as
$$L = \frac{1}{N_s} \sum_{(\mathbf{X}_i, \mathbf{Y}_i) \sim \mathcal{S}} L_{pce}\left(\mathbf{Y}_i', \mathbf{Y}_i\right) + \hat{\mathbf{Y}}_i L_{ce}(\mathbf{Y}_i', \tilde{\mathbf{M}}_i) + \hat{\mathbf{Y}}_i L_{IoU}(\mathbf{Y}_i', \tilde{\mathbf{M}}_i). \tag{11}$$
Where the first term is the partial cross-entropy loss $L_{pce}$ used to ensure consistency between the prediction maps and the sparse annotations $\mathbf{Y}_i$ [34]. The second and third terms are the cross-entropy loss $L_{ce}$ and the intersection-over-union loss $L_{IoU}$, both calculated using the pseudo label $\hat{\mathbf{Y}}_i$ [35].

# 4 Experiments

## 4.1 Experimental Setup

**Implementation details**. The image encoder uses ResNet50 as the backbone and is pre-trained on ImageNet [39]. The batch size is 36 and the learning rate is initialized as 0.0001, decreased by 0.1 every 80 epochs. For scribble supervision, we propose a nine-box strategy, namely constructing

Table 1: Results on COD with point supervision and scribble supervision. SCOD+ indicates integrating SCOD with our WS-SAM framework. The best two results are in **red** and **blue** fonts.

| Methods | Pub. | CHAMELEON | | | | CAMO | | | | COD10K | | | | NC4K | | | |
|---|---|---|---|---|---|---|---|---|---|---|---|---|---|---|---|---|---|
| | | $M\downarrow$ | $F_\beta\uparrow$ | $E_\phi\uparrow$ | $S_\alpha\uparrow$ | $M\downarrow$ | $F_\beta\uparrow$ | $E_\phi\uparrow$ | $S_\alpha\uparrow$ | $M\downarrow$ | $F_\beta\uparrow$ | $E_\phi\uparrow$ | $S_\alpha\uparrow$ | $M\downarrow$ | $F_\beta\uparrow$ | $E_\phi\uparrow$ | $S_\alpha\uparrow$ |
| Scribble Supervision | | | | | | | | | | | | | | | | | |
| SAM [19] | — | 0.207 | 0.595 | 0.647 | 0.635 | 0.160 | 0.597 | 0.639 | 0.643 | 0.093 | 0.673 | 0.737 | 0.730 | 0.118 | 0.675 | 0.723 | 0.717 |
| SAM-S [19] | — | 0.076 | 0.729 | 0.820 | 0.650 | 0.105 | 0.682 | 0.774 | 0.731 | 0.046 | 0.695 | 0.828 | 0.772 | 0.071 | 0.747 | 0.832 | 0.763 |
| WSSA [36] | CVPR20 | 0.067 | 0.692 | 0.860 | 0.782 | 0.118 | 0.615 | 0.786 | 0.696 | 0.071 | 0.536 | 0.770 | 0.684 | 0.091 | 0.657 | 0.779 | 0.761 |
| SCWS [37] | AAAI21 | 0.053 | 0.758 | 0.881 | 0.792 | 0.102 | 0.658 | 0.795 | 0.713 | 0.055 | 0.602 | 0.805 | 0.710 | 0.073 | 0.723 | 0.814 | 0.784 |
| TEL [38] | CVPR22 | 0.073 | 0.708 | 0.827 | 0.785 | 0.104 | 0.681 | 0.797 | 0.717 | 0.057 | 0.633 | 0.826 | 0.724 | 0.075 | 0.754 | 0.832 | 0.782 |
| SCOD [30] | AAAI23 | 0.046 | 0.791 | 0.897 | 0.818 | 0.092 | 0.709 | 0.815 | 0.735 | 0.049 | 0.637 | 0.832 | 0.733 | 0.064 | 0.751 | 0.853 | 0.779 |
| SCOD+ | — | 0.046 | 0.797 | 0.900 | 0.820 | 0.090 | 0.716 | 0.818 | 0.741 | 0.047 | 0.650 | 0.845 | 0.742 | 0.060 | 0.766 | 0.862 | 0.785 |
| Ours | — | 0.046 | 0.777 | 0.897 | 0.824 | 0.092 | 0.742 | 0.818 | 0.759 | 0.038 | 0.719 | 0.878 | 0.803 | 0.052 | 0.802 | 0.886 | 0.829 |
| Point Supervision | | | | | | | | | | | | | | | | | |
| SAM [19] | — | 0.207 | 0.595 | 0.647 | 0.635 | 0.160 | 0.597 | 0.639 | 0.643 | 0.093 | 0.673 | 0.737 | 0.730 | 0.118 | 0.675 | 0.723 | 0.717 |
| SAM-P [19] | — | 0.101 | 0.696 | 0.745 | 0.697 | 0.123 | 0.649 | 0.693 | 0.677 | 0.069 | 0.694 | 0.796 | 0.765 | 0.082 | 0.728 | 0.786 | 0.776 |
| WSSA [36] | CVPR20 | 0.105 | 0.660 | 0.712 | 0.711 | 0.148 | 0.607 | 0.652 | 0.649 | 0.087 | 0.509 | 0.733 | 0.642 | 0.104 | 0.688 | 0.756 | 0.743 |
| SCWS [37] | AAAI21 | 0.097 | 0.684 | 0.739 | 0.714 | 0.142 | 0.624 | 0.672 | 0.687 | 0.082 | 0.593 | 0.777 | 0.738 | 0.098 | 0.695 | 0.767 | 0.754 |
| TEL [38] | CVPR22 | 0.094 | 0.712 | 0.751 | 0.746 | 0.133 | 0.662 | 0.674 | 0.645 | 0.063 | 0.623 | 0.803 | 0.727 | 0.085 | 0.725 | 0.795 | 0.766 |
| SCOD [30] | AAAI23 | 0.092 | 0.688 | 0.746 | 0.725 | 0.137 | 0.629 | 0.688 | 0.663 | 0.060 | 0.607 | 0.802 | 0.711 | 0.080 | 0.744 | 0.796 | 0.758 |
| SCOD+ | — | 0.089 | 0.704 | 0.757 | 0.731 | 0.129 | 0.642 | 0.693 | 0.666 | 0.058 | 0.618 | 0.812 | 0.719 | 0.075 | 0.767 | 0.825 | 0.771 |
| Ours | — | 0.056 | 0.767 | 0.868 | 0.805 | 0.102 | 0.703 | 0.757 | 0.718 | 0.039 | 0.698 | 0.856 | 0.790 | 0.057 | 0.801 | 0.859 | 0.813 |

Table 2: Results for PIS and TOD with point supervision.

| Methods | Polyp Image Segmentation (PIS) | | | | | | | | | | | | Transparant Object Detection (TOD) | | | | | | | |
|---|---|---|---|---|---|---|---|---|---|---|---|---|---|---|---|---|---|---|---|---|
| | CVC-ColonDB | | | | ETIS | | | | Kvasir | | | | GDD | | | | GSD | | | |
| | $M\downarrow$ | $F_\beta\uparrow$ | $E_\phi\uparrow$ | $S_\alpha\uparrow$ | $M\downarrow$ | $F_\beta\uparrow$ | $E_\phi\uparrow$ | $S_\alpha\uparrow$ | $M\downarrow$ | $F_\beta\uparrow$ | $E_\phi\uparrow$ | $S_\alpha\uparrow$ | $M\downarrow$ | $F_\beta\uparrow$ | $E_\phi\uparrow$ | $S_\alpha\uparrow$ | $M\downarrow$ | $F_\beta\uparrow$ | $E_\phi\uparrow$ | $S_\alpha\uparrow$ |
| SAM [19] | 0.479 | 0.343 | 0.419 | 0.427 | 0.429 | 0.439 | 0.512 | 0.503 | 0.320 | 0.545 | 0.564 | 0.582 | 0.245 | 0.512 | 0.530 | 0.551 | 0.266 | 0.473 | 0.501 | 0.514 |
| SAM-P [19] | 0.194 | 0.587 | 0.664 | 0.671 | 0.144 | 0.625 | 0.719 | 0.715 | 0.108 | 0.793 | 0.811 | 0.802 | 0.164 | 0.668 | 0.715 | 0.625 | 0.177 | 0.687 | 0.730 | 0.668 |
| WSSA [36] | 0.127 | 0.645 | 0.732 | 0.713 | 0.123 | 0.647 | 0.733 | 0.762 | 0.082 | 0.822 | 0.852 | 0.828 | 0.173 | 0.652 | 0.710 | 0.616 | 0.185 | 0.661 | 0.712 | 0.650 |
| SCWS [37] | 0.082 | 0.674 | 0.758 | 0.787 | 0.085 | 0.646 | 0.768 | 0.731 | 0.078 | 0.837 | 0.860 | 0.831 | 0.170 | 0.631 | 0.702 | 0.613 | 0.172 | 0.706 | 0.738 | 0.673 |
| TEL [38] | 0.089 | 0.669 | 0.743 | 0.761 | 0.083 | 0.639 | 0.776 | 0.726 | 0.091 | 0.810 | 0.826 | 0.804 | 0.230 | 0.640 | 0.586 | 0.536 | 0.275 | 0.571 | 0.501 | 0.495 |
| SCOD [30] | 0.077 | 0.691 | 0.795 | 0.802 | 0.071 | 0.646 | 0.802 | 0.766 | 0.071 | 0.853 | 0.877 | 0.836 | 0.146 | 0.801 | 0.778 | 0.723 | 0.154 | 0.743 | 0.751 | 0.710 |
| SCOD+ | 0.074 | 0.702 | 0.806 | 0.803 | 0.066 | 0.670 | 0.811 | 0.769 | 0.068 | 0.860 | 0.880 | 0.836 | 0.129 | 0.818 | 0.796 | 0.732 | 0.145 | 0.761 | 0.765 | 0.720 |
| Ours | 0.043 | 0.721 | 0.839 | 0.816 | 0.037 | 0.694 | 0.849 | 0.797 | 0.046 | 0.878 | 0.917 | 0.877 | 0.078 | 0.858 | 0.863 | 0.775 | 0.089 | 0.839 | 0.841 | 0.764 |

the minimum outer wrapping rectangle of the foreground/background scribble and dividing it into a nine-box grid, to sample one point in each box and send them to SAM for segmentation mask generation. Following [2], all images are resized as $352 \times 352$ in both the training and testing phases. For SAM [19], we adopt the ViT-H SAM model to generate segmentation masks. We implement our method with PyTorch and run experiments on two RTX3090 GPUs.

**Baselines**. We explore SAM [19] for the WSCOS task by generating segmentation masks with sparse annotations as prompts and using the segmentation masks to train a COS segmenter. However, a more straightforward way to explore SAM for this task is to use the sparse annotation to fine-tune SAM and then directly apply SAM for the test. To verify the advantages of our method over this direct way, we construct two baseline methods, SAM-S and SAM-P, which fine-tune the mask decoder of SAM with scribble and point supervisions, respectively, by the partial cross-entropy loss. We will show the results of these two baselines in our comparative evaluations. For reference, we also report the results of the vanilla SAM. When applying SAM and its variants, SAM-S and SAM-P, on test images, we use the automatic prompt generation strategy and report the results with the highest IoU scores.

**Metrics**. Following existing methods [1, 2], we use four common metrics for evaluation, including mean absolute error ($M$), adaptive F-measure ($F_\beta$) [40], mean E-measure ($E_\phi$) [41], and structure measure ($S_\alpha$) [42]. Smaller $M$, or larger $F_\beta$, $E_\phi$, $S_\alpha$ means better segmentation performance.

## 4.2 Comparative Evaluation

We perform evaluations on the following COS tasks, namely, Camouflaged Object Detection, Polyp Image Segmentation (PIS), and Transparent Object Detection (TOD). For all the tasks, we evaluate the performance with point annotations. We follow the previous weakly-supervised segmentation method [43] and randomly select two points (one from the foreground and one from the background) from the training masks as the point annotations. For COD, we additionally evaluate the performance using scribble annotations, using the scribble data provided in [43].

**Camouflaged object detection**. Four datasets are used for experiments, *i.e.*, CHAMELEON [44], CAMO [45], COD10K [1], and NC4K [16]. Table 1 shows that our method reaches the best performance over all competing methods and baselines. Notably, while SAM has shown impressive

Table 3: Ablations for WS-SAM.

| Baseline | MAF | PLW | ILS | $M\downarrow$ | $F_\beta\uparrow$ | $E_\phi\uparrow$ | $S_\alpha\uparrow$ |
|---|---|---|---|---|---|---|---|
| ✓ | | | | 0.052 | 0.674 | 0.838 | 0.737 |
| ✓ | ✓ | | | 0.047 | 0.689 | 0.853 | 0.772 |
| ✓ | ✓ | ✓ | | 0.044 | 0.697 | 0.866 | 0.793 |
| ✓ | ✓ | ✓ | ✓ | **0.038** | **0.719** | **0.878** | **0.803** |

Table 4: Ablations for MFG.

| Metrics | w/o MFG | FG->SA | w/o multiscale | WGM->FC | w/ MFG |
|---|---|---|---|---|---|
| $M\downarrow$ | 0.044 | 0.038 | 0.040 | 0.039 | **0.038** |
| $F_\beta\uparrow$ | 0.684 | 0.708 | 0.702 | 0.710 | **0.719** |
| $E_\phi\uparrow$ | 0.857 | 0.868 | 0.858 | 0.871 | **0.878** |
| $S_\alpha\uparrow$ | 0.780 | 0.797 | 0.783 | 0.792 | **0.803** |

Table 5: Results of MFG with full supervision.

| Metrics | Baseline | SegMaR [17] | PreyNet [8] | FGANet [15] | Ours |
|---|---|---|---|---|---|
| $M\downarrow$ | 0.035 | 0.035 | 0.034 | **0.032** | **0.032** |
| $F_\beta\uparrow$ | 0.688 | 0.699 | **0.715** | 0.708 | 0.706 |
| $E_\phi\uparrow$ | 0.879 | 0.890 | 0.894 | 0.894 | **0.897** |
| $S_\alpha\uparrow$ | 0.812 | **0.813** | **0.813** | 0.803 | **0.813** |

Table 6: Results on multi-object images.

| Metrics | SCWS [37] | TEL [38] | SCOD [30] | Ours |
|---|---|---|---|---|
| $M\downarrow$ | 0.094 | 0.101 | 0.084 | **0.070** |
| $F_\beta\uparrow$ | 0.378 | 0.350 | 0.381 | **0.452** |
| $E_\phi\uparrow$ | 0.740 | 0.726 | 0.718 | **0.772** |
| $S_\alpha\uparrow$ | 0.625 | 0.617 | 0.643 | **0.687** |

performance for natural scene images, its performance on the challenging COD task is far from the existing methods particularly designed for this task. We do see performance gains after finetuning SAM with point (SAM-P) and scribble (SAM-S) supervision, but the results are still far below our method. This substantiates the superiority of our way of leveraging SAM to generate segmentation masks with sparse annotations and use the segmentation masks to train the segmenter. To verify our performance improvement over the existing WSCOS methods does not merely come from the usage of SAM, we integrate the most recent WSCOS method, SCOD [30], into our WS-SAM framework to also leverage the additional mask supervision. This results in the method, "SCOD+". We can see that our method still shows better performance, further verifying our advantages for this task.

**Polyp image segmentation**. Three widely-used Polyp datasets are selected, namely *CVC-ColonDB* [46], *ETIS* [47], and *Kvasir* [48]. Table 2 shows that our method significantly surpasses the second-best method, SCOD, with point supervision. SAM and SAM-P do not perform well on this task, further substantiating their weakness on this challenging segmentation task. While empowering SCOD with the proposed WS-SAM framework indeed improves the performance, the results are still lower than our method. This again verifies our benefit in handling challenging segmentation tasks.

**Transparent object detection**. Two datasets, GDD [6] and GSD [29], are used for evaluation. As shown in Table 2, our method surpasses all baseline methods and existing methods for this task as well. This shows strong robustness and generalizability of our proposed method.

## 4.3 Ablation Study

Our method includes two main components, the SAM-based weakly-supervised mask generation framework, WS-SAM, and the multi-scale feature grouping (MFG) module. We conduct ablation studies about these two components on *COD10K* of the COD task with scribble supervision.

**Ablation study for WS-SAM**. We establish a baseline by using SAM to generate only one segmentation mask from one training image without augmentations for model training. On top of this baseline, we add the multi-augmentation fusion (MAF), pixel-level weighting (PLW), and image-level selection (ILS) techniques. Table 3 showing adding these components helps improve the performance, thus demonstrating their effectiveness.

**Ablation study for MFG**. We examine the effect of MFG by completely removing the MFG module, substituting the proposed feature grouping (FG) with slot attention (SA) [33], removing the multi-scale strategy, and substituting the weighted gate mechanism (WGM) with fixed coefficient (FC). Table 4 shows that our designs reach better performance than the alternative ones.

## 4.4 Further Analysis

**MFG for the *fully-supervised* setting**. The proposed MFG module is designed to evacuate discriminative cues from concealed images. We have demonstrated its effectiveness with sparse annotations for the weakly-supervised setting. However, it is expected to also work in the fully-supervised setting. To verify this, we conduct experiments for the standard fully-supervised COD task. Table 5 shows the results on the *COD10K* dataset. We can see that MFG indeed helps improve the performance of the baseline model, to the level comparative with state-of-the-art methods.

**Performance on multi-object images**. The proposed MFG module evacuates discriminative cues by performing feature grouping at different granularities, which facilitates discovering multiple objects

Table 7: Parameter analysis on $K$, $\tau_a$, $\tau_r$, $T$, and $(N_1, N_2)$.

| Metrics | $K$ | | | | $\tau_a$ | | | | $\tau_r$ | | | | $T$ | | | | $(N_1, N_2)$ | | | |
|---|---|---|---|---|---|---|---|---|---|---|---|---|---|---|---|---|---|---|---|---|
| | 1 | 6 | 12 | 18 | 0.05 | 0.1 | 0.2 | 0.3 | 0.3 | 0.5 | 0.7 | 0.9 | 1 | 2 | 3 | 4 | (2,4) | (2,8) | (4,8) | (2,4,8) |
| $M \downarrow$ | 0.052 | 0.042 | **0.038** | 0.039 | **0.037** | 0.038 | 0.038 | 0.040 | **0.038** | 0.038 | 0.039 | 0.040 | 0.039 | 0.039 | **0.038** | 0.038 | **0.038** | 0.038 | 0.039 | **0.038** |
| $F_\beta \uparrow$ | 0.674 | 0.697 | **0.719** | 0.718 | 0.706 | **0.719** | 0.716 | 0.704 | **0.723** | 0.719 | 0.715 | 0.700 | 0.706 | 0.715 | 0.719 | **0.720** | 0.719 | 0.714 | 0.711 | **0.721** |
| $E_\phi \uparrow$ | 0.838 | 0.857 | **0.878** | 0.878 | 0.868 | **0.878** | 0.876 | 0.865 | 0.866 | **0.878** | 0.874 | 0.851 | 0.862 | 0.872 | **0.878** | 0.876 | **0.878** | 0.875 | 0.873 | **0.878** |
| $S_\alpha \uparrow$ | 0.737 | 0.776 | **0.803** | 0.800 | 0.795 | 0.803 | **0.805** | 0.793 | 0.792 | **0.803** | 0.789 | 0.781 | 0.794 | 0.800 | 0.803 | **0.805** | **0.803** | 0.802 | 0.799 | 0.802 |

Figure 4: Visualized results for COD tasks.

in images. To verify this, we evaluate the performance on the 186 images with more than one object from $COD10K$. Table 6 shows that MFG achieves the best performance, surpassing the second-best method (SCOD) by $10.7\%$. This gap is large than that with all test images, where the gap is $5.8\%$.

**Randomness of point supervision.** We follow the existing point-supervision segmentation methods and randomly select points from ground truth masks as the point annotation. To study the variance of the random selection, we repeat the random selection 5 times and calculate the mean and standard deviation of the results. Fig. 5 shows that our method reaches the best results while having the smallest deviation.

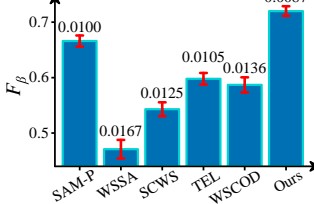

Figure 5: Five runs results with varied point annotations.

**Number of augmented views** $K$. Table 7 shows that more augmented views help improve performance in the beginning, but the effect turns weaker when further increasing it.

**Hyperparameters in image-level selection**. Table 7 shows that it is best to set the absolute uncertainty threshold $\tau_a = 0.1$ and the relative uncertainty threshold $\tau_r = 0.5$, and our method is not sensitive to these two parameters.

**Hyperparameters in MFG**. Table 7 shows that MFG achieves the best results when the iteration number $T$ is set as 3, and the groups and scales setting is set as $(N_1, N_2) = (2, 4)$. Notice that when adopting $(2, 4, 8)$, RK2 is replaced with the third-order RK structure, resulting in extra computational burden with limited benefits. Hence, we select the RK2 structure with $(N_1, N_2) = (2, 4)$.

**Result visualization.** Fig. 4 shows the prediction maps with point supervision. We can see that our method produces more complete results than existing methods and localizes multiple objects more comprehensively. More visualization results can be found in the supplementary materials.

## 5 Conclusions

This paper proposes a new WSCOS method that includes two key components. The first one is the WS-SAM framework that generates segmentation masks with the recently proposed vision foundation model, SAM, and proposes multi-augmentation result fusion, pixel-level uncertainty weighting, and image-level uncertainty filtration to get reliable pseudo labels to train a segmentation model. The second is the MFG module that leverages the extracted clues for additional nuanced discrimination information. MFG improves feature coherence from a grouping aspect, allowing for alleviating incomplete segmentation and better multiple-object segmentation. Experiments on multiple WSCOS tasks confirm the superiority of our method over the baseline and existing methods.

**Acknowledgements:** This research is partly supported by the National Key R&D Program of China (Grants No. 2020AAA0108302 & 2020AAA0108303), and Shenzhen Science and Technology Project (Grant No. JCYJ20200109143041798) & Shenzhen Stable Supporting Program (WDZC20200820200655001) & Shenzhen Key Laboratory of next generation interactive media innovative technology (Grant No. ZDSYS20210623092001004). The authors express their appreciation to Dr. Fengyang Xiao for her insightful comments, improving the quality of this paper.

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
