# Supplementary Materials for
# Weakly-Supervised Concealed Object Segmentation with SAM-based Pseudo Labeling and Multi-scale Feature Grouping

**Chunming He**[1]*, **Kai Li**[2]*, **Yachao Zhang**[1] , **Guoxia Xu**[3] ,
**Longxiang Tang**[1] , **Yulun Zhang**[4] , **Zhenhua Guo**[5] , and **Xiu Li**[1]†
[1]Shenzhen International Graduate School, Tsinghua University, [2]NEC Laboratories America,
[3]Nanjing University of Posts and Telecommunications, [4]ETH Zürich, [5]Tianyi Traffic Technology

## Contents

## A   Segmenter

In the manuscript, we only present the proposed multi-scale feature grouping (MFG) module for space limitation. In this section, we further describe in detail the structure of the novel segmenter (see Fig. 1), which integrates MFG with the encoder and decoder architecture utilized in FEDER [1].

**Encoder**. Following [1], we employ ResNet50 [2] as the basic encoder $E$. Given a concealed image $\mathbf{X}_i$ with the size of $H \times W$, we start by using $E$ to encode $\mathbf{X}_i$ into a set of features $\{\tilde{\mathbf{F}}_i^k\}_{k=0}^4$ with the resolution of $\frac{H}{2^{k+1}} \times \frac{W}{2^{k+1}}$. Afterward, we follow the practice in [1] and cascade R-Net [3] $R$ to the basic encoder to transform $\{\tilde{\mathbf{F}}_i^k\}_{k=1}^4$ to $\{\mathbf{F}_i^k\}_{k=1}^4$ for channel reduction, which is a more compact feature space with the channel number of 64. Besides, given that $\tilde{\mathbf{F}}_i^4$ is rich in semantic information, we sent it to an atrous spatial pyramid pooling (ASPP) [4] $A_s$ to generate a coarse prediction map $\tilde{\mathbf{Y}}_i^5$ to guide the segmentation: $\tilde{\mathbf{Y}}_i^5 = A_s(\tilde{\mathbf{F}}_i^4)$.

**MFG**. To balance performance and efficiency, we only apply MFG to the bottom layer and get $\dot{\mathbf{F}}_i^4$, because the semantic information enriched by the deep features in the bottom layer is most conducive to MFG to mine the feature coherence. To aggregate the cross-layer features, we refer to the practice in [1] and employ the joint attention module $JA(\cdot)$, which consists of spatial attention [5] and channel

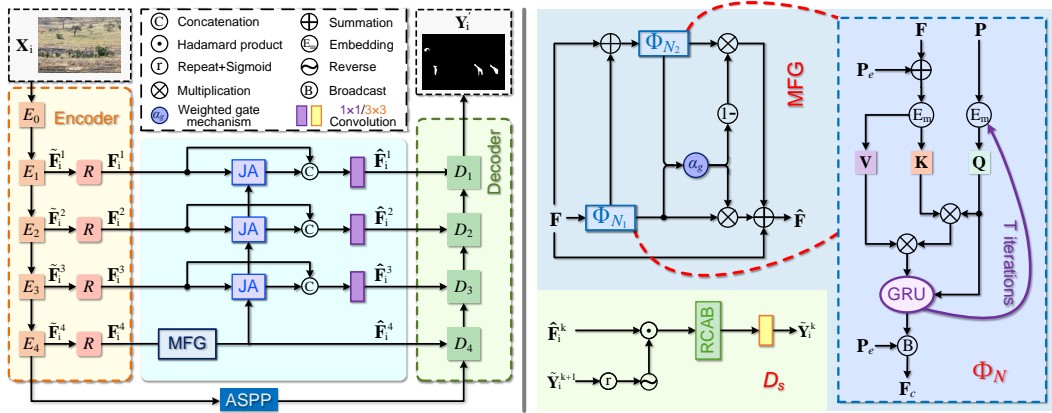

Figure 1: Framework of the proposed segmenter with MFG.

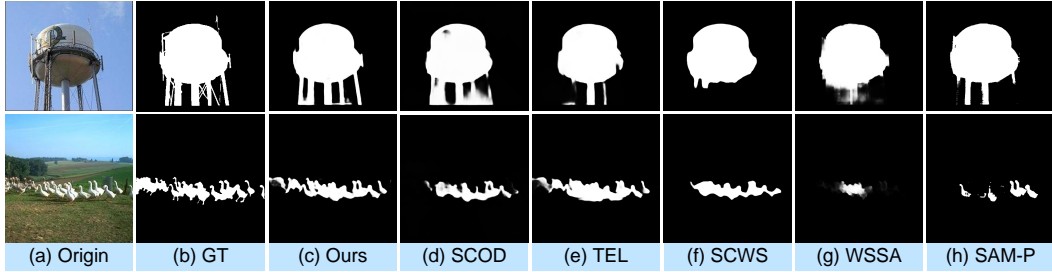

| (a) Origin | (b) GT | (c) Ours | (d) SCOD | (e) TEL | (f) SCWS | (g) WSSA | (h) SAM-P |

Figure 2: Qualitative analysis of MFG and other five SOTAs on SOD with point supervision.

attention [6]:

$$\ddot{\mathbf{F}}_i^k = JA(\mathbf{F}_i^k, up(\ddot{\mathbf{F}}_i^{k+1})), \tag{1}$$

where $\ddot{\mathbf{F}}_i^4 = \dot{\mathbf{F}}_i^4$ and $up(\cdot)$ denotes the up-sampling operation. Having combined the cross-layer features, we further define the latent features $\{\hat{\mathbf{F}}_i^k\}_{k=1}^3$ conveyed to the decoder:

$$\hat{\mathbf{F}}_i^k = conv1(cat(\mathbf{F}_i^k, \ddot{\mathbf{F}}_i^k)), \tag{2}$$

where $cat(\cdot)$ denote the concatenation operation and $conv1(\cdot)$ means $1 \times 1$ convolution. $\hat{\mathbf{F}}_i^4 = \dot{\mathbf{F}}_i^4$.

**Decoder**. The prediction maps of segmenters trained with sparse annotations often contain low-confidence ambiguous regions due to the complex scenes of concealed objects and the limited discriminative capacity of the segmenter. To address this issue, we follow the architecture of the decoder used in FEDER [1], which utilizes the prediction map of the previous decoder to excavate cues from the low-confidence regions. By doing so, the segmenter can more effectively detect the undetected parts in these regions and thus improve the segmentation performance. Therefore, the prediction map $\{\tilde{\mathbf{Y}}_i^k\}_{k=1}^4$ is defined as follows:

$$\tilde{\mathbf{Y}}_i^k = conv3(RCAB(\hat{\mathbf{F}}_i^k \odot rv(S(rp(\tilde{\mathbf{Y}}_i^{k+1}))))), \tag{3}$$

where $rp(\cdot)$, $S(\cdot)$, $rv(\cdot)$, $\odot$, and $conv3(\cdot)$ denote repeat, Sigmoid, reverse (element-wise subtraction with 1), Hadamard product, and $3 \times 3$ convolution. $RCAB(\cdot)$ is the residual channel attention block [7] and we employ this block to emphasize the noteworthy information. The final segmentation result $\mathbf{Y}_i'$ is $\tilde{\mathbf{Y}}_i^1$.

# B  Experiments

## B.1  Weakly-supervised Salient Object Detection

In the manuscript, we have verified the superiority of our method in weakly-supervised concealed object segmentation. In this section, we further validate our performance on weakly-supervised salient object detection.

**Quantitative analysis**. We employ S-DUTS [9] and P-DUTS [13] to train the methods with scribble and point supervisions and evaluate the segmentation performance on five common benchmarks, *i.e.*, *DUT-OMRON* [14], *DUTS-test* [15], *ECSSD* [16], *HKU-IS* [17], and *PASCAL-S* [18].

Table 1: Quantitative comparisons of our method and other 5 ResNet50-based SOTAs on the SOD task with scribble and point supervisions. The best two results are in red and blue fonts.

| Methods | DUT-OMRON | | | | DUTS-test | | | | ECSSD | | | | HKU-IS | | | | PASCAL-S | | | |
|---|---|---|---|---|---|---|---|---|---|---|---|---|---|---|---|---|---|---|---|---|
| | $M\downarrow$ | $F_\beta\uparrow$ | $E_\phi\uparrow$ | $S_\alpha\uparrow$ | $M\downarrow$ | $F_\beta\uparrow$ | $E_\phi\uparrow$ | $S_\alpha\uparrow$ | $M\downarrow$ | $F_\beta\uparrow$ | $E_\phi\uparrow$ | $S_\alpha\uparrow$ | $M\downarrow$ | $F_\beta\uparrow$ | $E_\phi\uparrow$ | $S_\alpha\uparrow$ | $M\downarrow$ | $F_\beta\uparrow$ | $E_\phi\uparrow$ | $S_\alpha\uparrow$ |
| Scribble Supervision | | | | | | | | | | | | | | | | | | | | |
| SAM-F [8] | 0.077 | 0.696 | 0.729 | 0.734 | 0.103 | 0.713 | 0.741 | 0.737 | 0.103 | 0.728 | 0.741 | 0.744 | 0.119 | 0.651 | 0.657 | 0.657 | 0.116 | 0.695 | 0.716 | 0.712 |
| SAM-S [8] | 0.060 | 0.772 | 0.868 | 0.822 | 0.048 | 0.817 | 0.845 | 0.783 | 0.062 | 0.834 | 0.886 | 0.871 | 0.051 | 0.847 | 0.919 | 0.803 | 0.072 | 0.782 | 0.793 | 0.795 |
| WSSA [9] | 0.068 | 0.713 | 0.840 | 0.785 | 0.062 | 0.747 | 0.857 | 0.803 | 0.059 | 0.865 | 0.901 | 0.865 | 0.047 | 0.858 | 0.927 | 0.865 | 0.092 | 0.788 | 0.831 | 0.791 |
| SCWS [10] | 0.060 | 0.764 | 0.862 | 0.812 | 0.049 | 0.823 | 0.890 | 0.841 | 0.049 | 0.900 | 0.908 | 0.882 | 0.038 | 0.896 | 0.938 | 0.882 | 0.077 | 0.823 | 0.846 | 0.813 |
| TEL [11] | 0.058 | 0.753 | 0.864 | 0.818 | 0.045 | 0.842 | 0.901 | 0.863 | 0.039 | 0.921 | 0.935 | 0.907 | 0.033 | 0.934 | 0.944 | 0.906 | 0.065 | 0.834 | 0.883 | 0.858 |
| SCOD [12] | 0.061 | 0.762 | 0.865 | 0.814 | 0.046 | 0.830 | 0.892 | 0.855 | 0.042 | 0.917 | 0.922 | 0.910 | 0.035 | 0.933 | 0.942 | 0.900 | 0.069 | 0.828 | 0.864 | 0.847 |
| SCOD+ | 0.057 | 0.774 | 0.872 | 0.822 | 0.042 | 0.841 | 0.907 | 0.862 | 0.038 | 0.930 | 0.935 | 0.917 | 0.032 | 0.935 | 0.947 | 0.906 | 0.062 | 0.838 | 0.881 | 0.855 |
| Ours | 0.050 | 0.798 | 0.885 | 0.844 | 0.034 | 0.870 | 0.931 | 0.890 | 0.032 | 0.942 | 0.955 | 0.926 | 0.026 | 0.935 | 0.958 | 0.922 | 0.055 | 0.869 | 0.914 | 0.871 |
| Point Supervision | | | | | | | | | | | | | | | | | | | | |
| SAM-F [8] | 0.077 | 0.696 | 0.729 | 0.734 | 0.103 | 0.713 | 0.741 | 0.737 | 0.103 | 0.728 | 0.741 | 0.744 | 0.119 | 0.651 | 0.657 | 0.657 | 0.116 | 0.695 | 0.716 | 0.712 |
| SAM-P [8] | 0.075 | 0.713 | 0.738 | 0.759 | 0.085 | 0.768 | 0.782 | 0.764 | 0.090 | 0.787 | 0.785 | 0.776 | 0.074 | 0.734 | 0.704 | 0.698 | 0.102 | 0.735 | 0.738 | 0.730 |
| WSSA [9] | 0.082 | 0.690 | 0.727 | 0.708 | 0.085 | 0.742 | 0.756 | 0.733 | 0.087 | 0.824 | 0.831 | 0.806 | 0.080 | 0.786 | 0.715 | 0.718 | 0.104 | 0.733 | 0.720 | 0.712 |
| SCWS [10] | 0.076 | 0.668 | 0.720 | 0.720 | 0.080 | 0.747 | 0.769 | 0.752 | 0.089 | 0.815 | 0.828 | 0.797 | 0.076 | 0.801 | 0.773 | 0.738 | 0.098 | 0.752 | 0.742 | 0.733 |
| TEL [11] | 0.073 | 0.699 | 0.736 | 0.717 | 0.074 | 0.758 | 0.775 | 0.761 | 0.078 | 0.857 | 0.840 | 0.813 | 0.071 | 0.817 | 0.785 | 0.764 | 0.093 | 0.761 | 0.760 | 0.748 |
| SCOD [12] | 0.078 | 0.687 | 0.729 | 0.705 | 0.079 | 0.741 | 0.769 | 0.739 | 0.082 | 0.834 | 0.809 | 0.792 | 0.074 | 0.808 | 0.769 | 0.753 | 0.097 | 0.749 | 0.733 | 0.729 |
| SCOD+ | 0.075 | 0.694 | 0.734 | 0.712 | 0.076 | 0.766 | 0.778 | 0.747 | 0.080 | 0.851 | 0.820 | 0.800 | 0.071 | 0.816 | 0.781 | 0.758 | 0.094 | 0.758 | 0.747 | 0.741 |
| Ours | 0.064 | 0.741 | 0.828 | 0.790 | 0.052 | 0.821 | 0.880 | 0.837 | 0.048 | 0.910 | 0.920 | 0.891 | 0.048 | 0.884 | 0.903 | 0.863 | 0.075 | 0.814 | 0.865 | 0.824 |

Figure 3: Visualized results for the three WSCOS tasks with point supervision.

(a) Origin  (b) GT  (c) Ours  (d) SCOD  (e) TEL  (f) SCWS  (g) WSSA  (h) SAM-P

As depicted in Table 1, the proposed MFG achieves the leading place in all benchmarks and our WS-SAM framework enhances the segmentation performance of SCOD in 3.0% and 1.9% with scribble and point supervision, which demonstrates the advancement of our MFG and WS-SAM.

**Qualitative analysis**. In Fig. 1, MFG can better ensure the integrity and comprehensiveness of the prediction maps, which demonstrates the superiority of the proposed WS-SAM framework and MFG model. Notice that the effectiveness of the learnable SAM, *i.e.*, SAM-S, is significantly improved in the SOD task. This is likely due to the fact that SOD typically has more obvious discrimination features compared to the COS tasks, making it easier to obtain performance gains through fine-tuning.

## B.2 Weakly-supervised Concealed Object Segmentation

**Result visualization.** We provide more result visualizations of the weakly-supervised concealed object segmentation. Fig. 3 shows the prediction maps with point supervision. As exhibited in Fig. 3, we discover that our technique produces more complete results than existing segmentors and localizes multiple objects more comprehensively. This advancement performance is attributed not only to the accurate pseudo-labels generated by our WS-SAM framework, which improves the discrimination capacity of the segmenter, but also to the proposed MFG module that can adaptively

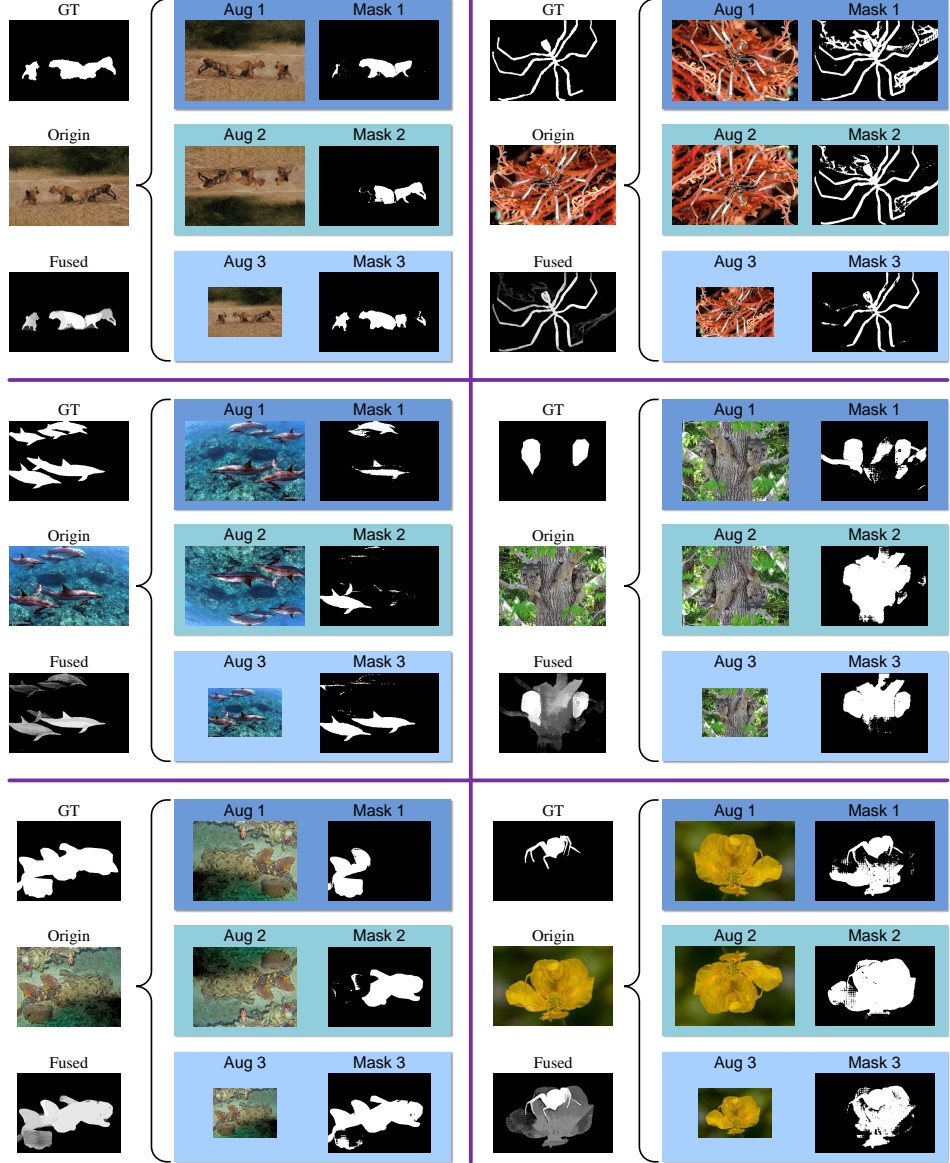

Figure 4: Masks of SAM with different augmented images. We inversely transform the masks to keep consistent with the original image. It is observed that fused masks contain more accurate and complete segmentation information.

enhance feature coherence from a clustering aspect, thus alleviating incomplete segmentation and facilitating multi-object segmentation.

## B.3 Ablation Study and Analysis

**Visualization of masks of different augmented images**. We provide more visualizations to explore the impact of data augmentations on segmentation. As exhibited in Fig. 4, it is evident that the generated masks exhibit considerable shape variations. Despite undergoing image transformations, certain regions consistently exhibit overlapping predictions, which align with accurately predicted foreground regions by SAM. Furthermore, these masks demonstrate complementarity, wherein some foreground regions missed by one mask are captured by other masks, enhancing the overall coverage of the foreground objects. Therefore, it is desirable to randomly combine the masks of those augmented images for pseudo label generation, which highlights the noteworthy parts and ensures the comprehensiveness of the generated pseudo labels.

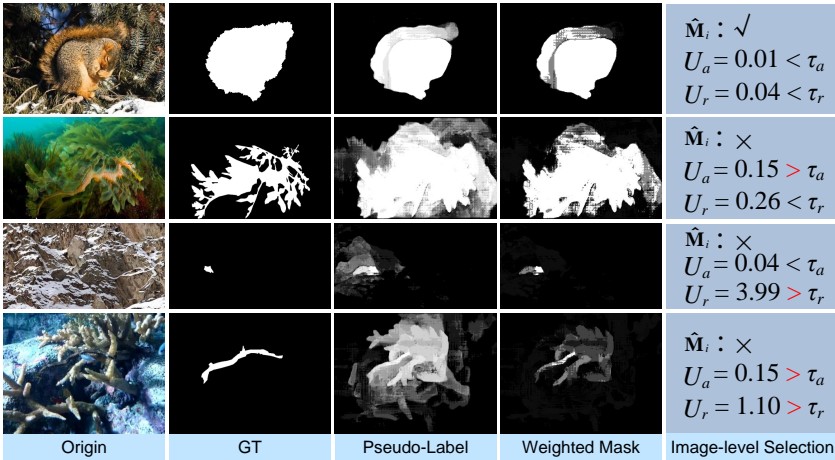

Figure 5: Visualization of pseudo-labels with pixel-level weighting and image-level selection. Weighted masks mean multiplying $\tilde{\mathbf{M}}_i$ and $1 - \tilde{\mathbf{E}}_i$, which assign higher weights to reliable pixels.

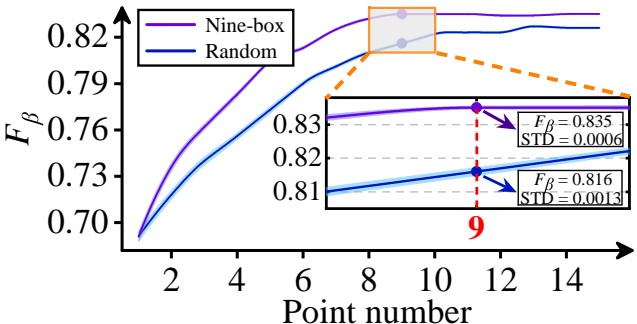

Figure 6: Parameter analysis on $C$.

**Visualization of filtered out pseudo-labels**. In Fig. 5, we present the visualization of low-quality pseudo-labels that are filtered out, along with the corresponding weighted masks. For comparison, we also provide a high-quality pseudo-label, which makes up the majority of the training set. As exhibited in Fig. 5, we observe that for images whose absolute uncertainty $U_a$ exceeds the threshold $\tau_a$, a large number of low-confidence regions often appear in the generated pseudo-labels. For those images whose relative uncertainty $U_r$ is higher than the threshold $\tau_r$, the proportion of low-confidence regions to the estimated foreground regions is often higher, particularly in small objects. At this point, the corresponding pixel-level mask may not accurately filter out the correct pixels because even those pixels in those complex images that SAM is very certain of in the pseudo-labels may still carry a risk of misclassification. Therefore, filtering out these images during training can improve the segmentation performance and generalization ability of the segmenter.

**Parameter analysis on point number** $C$. In the manuscript, for scribble supervision, we propose a nine-box strategy, namely constructing the minimum outer wrapping rectangle of the foreground/background scribble and dividing it into a nine-box grid, to sample one point in each box and send them to SAM for segmentation mask generation. In this section, we conduct an experiment on the training set of COD to investigate the effect of the number $C$ of key points sampled for scribble labels on the quality of the generated pseudo-labels. To ensure the comprehensiveness and validity of our findings, we perform ten trials and calculate the average $F_\beta$ score. Fig. 6 shows that the proposed nine-box strategy results in higher quality pseudo-labels compared to randomly selected points. Moreover, as the quality of the generated pseudo-labels becomes stable when $C = 9$, we set the point number $C$ to 9 in this paper. We also observe that our nine-box strategy is a highly stable approach with minimal standard deviation.

**Performance of MFG in different stages**. To balance performance and efficiency, we only apply MFG to the bottom layer and get $\dot{\mathbf{F}}_i^4$, because the semantic information enriched by the deep features in the bottom layer is most conducive to MFG to mine the feature coherence. We verify this argument by ablating the module in different stages. In Table 2, the notation $(*, *, *, *)$ means whether MFG

Table 2: Effect of the MFG module.

| Methods | COD10K | | | |
|---|---|---|---|---|
| | $M\downarrow$ | $F_\beta\uparrow$ | $E_\phi\uparrow$ | $S_\alpha\uparrow$ |
| (0,0,0,0) | 0.044 | 0.684 | 0.857 | 0.780 |
| (0,0,0,1) | 0.038 | 0.719 | 0.878 | **0.803** |
| (0,0,1,1) | 0.038 | 0.716 | 0.881 | 0.802 |
| (0,1,1,1) | 0.038 | 0.717 | **0.882** | 0.802 |
| (1,1,1,1) | **0.037** | **0.720** | 0.876 | 0.800 |

Table 3: Point-supervised results of MFG with different point annotation manners.

| Methods | COD10K | | | |
|---|---|---|---|---|
| | $M\downarrow$ | $F_\beta\uparrow$ | $E_\phi\uparrow$ | $S_\alpha\uparrow$ |
| Center | 0.040 | 0.726 | **0.858** | 0.791 |
| Edge1 | 0.043 | 0.709 | 0.850 | 0.786 |
| Edge2 | 0.043 | 0.713 | 0.847 | 0.787 |
| Random1 | **0.039** | 0.724 | 0.857 | **0.793** |
| Random2 | **0.039** | **0.729** | 0.854 | 0.785 |

Table 4: Ablations of WS-SAM with SAM-F.

| Baseline | MAF | PLW | ILS | $M\downarrow$ | $F_\beta\uparrow$ | $E_\phi\uparrow$ | $S_\alpha\uparrow$ |
|---|---|---|---|---|---|---|---|
| ✓ | | | | 0.052 | 0.674 | 0.838 | 0.737 |
| ✓ | ✓ | | | 0.047 | 0.689 | 0.853 | 0.772 |
| ✓ | ✓ | ✓ | | 0.044 | 0.697 | 0.866 | 0.793 |
| ✓ | ✓ | ✓ | ✓ | **0.038** | **0.719** | **0.878** | **0.803** |

Table 5: Ablations of WS-SAM with SAM-S.

| Baseline | MAF | PLW | ILS | $M\downarrow$ | $F_\beta\uparrow$ | $E_\phi\uparrow$ | $S_\alpha\uparrow$ |
|---|---|---|---|---|---|---|---|
| ✓ | | | | 0.049 | 0.685 | 0.847 | 0.743 |
| ✓ | ✓ | | | 0.046 | 0.692 | 0.858 | 0.780 |
| ✓ | ✓ | ✓ | | 0.043 | 0.701 | 0.870 | 0.795 |
| ✓ | ✓ | ✓ | ✓ | **0.039** | **0.722** | **0.875** | **0.802** |

is retained in the $s^{th}$ layer, *e.g.*, $(0,0,0,1)$ indicates MFG is retained in the $4^{th}$ stage. As shown in Table 2, MFG achieves promising performance when MFG is retained only in the $4^{th}$ stage, and additional MFG modules provide limited gains. Therefore, we select the version of $(0,0,0,1)$ to balance the performance and effectiveness.

**Impact of data annotation with point supervision**. Table 3 presents the detailed segmentation results of MFG under different point annotation manners, where we find small variations in segmentation results for the center and randomly selected points, further proving the stability of our WS-SAM framework. However, when the point is selected on the edge, the segmentation performance declines, as concealed objects with blurred edges often result in inaccurate pseudo-labels generated by SAM.

**Why not employ the fine-tuned SAM to generate pseudo-labels**? Fine-tuned versions of SAM, namely SAM-S and SAM-P, have demonstrated improved performance in the inference phase compared to the fixed SAM (SAM-F). However, as presented in Tables 4 and 5, the use of the pseudo-labels generated by SAM-S only brings limited improvements for MFG, compared to utilizing those labels generated by SAM-F. We then conduct detailed breakdown ablations to investigate the reasons. As illustrated in Tables 4 and 5, when using only baseline, WS-SAM with SAM-S achieves higher performance than that with SAM-F, which indicates the effectiveness of the utilized fine-tuning strategy. Nevertheless, such leadership is mitigated by the addition of the MAF and PLF mechanisms, demonstrating the advancement of our proposed MAF and PLF mechanisms. Furthermore, when further combining the ILS strategy, the version with SAM-F even has better performance on $COD10K$. This is not only attributed to the image filtration strategy of ILS but also indicates that being trained with the pseudo-labels generated by SAM-S can harm the generalization of the segmenter. Such harm may arise from the sparse annotations or limited data in the finetuning process. To sum up, we only use the fixed SAM in this paper, which has already achieved outstanding performance in the field of weakly-supervised COS. We leave the exploration of how to use a learnable SAM to further enhance the weakly-supervised COS task as a future work.

# C  Limitations and Future Works

## C.1  WS-SAM

In our future work, as mentioned above, we plan to design a more refined fine-tuning strategy for SAM to provide targeted changes for the intrinsic similarity of concealed objects [19], which will improve the quality of the pseudo-labels provided. Additionally, we will explore modifying the network architecture to enable real-time adjustment of SAM based on the feedback of the segmenter. This will allow the two components, namely learnable SAM and segmenter, to synergistically enhance each other during training. In addition, we also consider extending our framework to other settings, such as unsupervised learning [20] and semi-supervised learning [21].

## C.2 MFG

MFG heavily relies on the features already extracted by the segmenter for clustering. In this case, we aim to enhance the discriminative feature extraction capacity of the segmenter to ensure that the MFG module can better mine the discriminative cues of concealed objects. Moreover, we will consider utilizing our MFG to extract more semantic-level information [22–25] and thus extend this technique to be a plug-and-play strategy to empower more fields [26–29]. Additionally, we will consider using other self-excavation techniques to mine the valuable information [30–33] or incorporating more powerful architectures, e.g., dynamic networks [34–36], transformer [37, 38], and diffusion model [29, 39], with more strategic pretrain networks [40, 41]. Furthermore, it would be desirable to employ image quality assessment techniques [42–44] to distinguish hard samples, which helps to improve the quality of the generated pseudo labels.