# OpenReview forum: "Weakly-Supervised Concealed Object Segmentation with SAM-based Pseudo Labeling and Multi-scale Feature Grouping"
_NeurIPS.cc/2023/Conference — NeurIPS 2023 poster_

### Official Review · Reviewer_DmES · 2023-07-01

**Soundness:** 3 good
**Presentation:** 2 fair
**Contribution:** 2 fair
**Rating:** 7
**Confidence:** 4

**Summary:**

This paper proposes to tackle Concealed Object Segmentation tasks by generating high-quality pseudo segmentation masks using state-of-the-art vision foundation models--SAM.  SAM takes prompts as provided sparse annotations (e.g. points or scribbles) to produce figure-ground pixel labels, which are used to train the segmenter.  Yet, the task remains challenging as SAM is not robust to out-of-distribution images where objects visually blended with the surrounding environment.  This paper proposes to infer and select high-quality pseudo labels using 1) multi-augmentation result fusion and 2) entropy-based label filtering.  Additionally, this paper proposes a multi-scale feature grouping machinery to enhance the performance of the segmenter.  The proposed method is demonstrated on multiple benchmark datasets and achieves good results.



**Strengths:**

1. The motivation is elaborated well.  It's easy to understand what the tasks and the challenges are.
2. The performance is good among most of the benchmark datasets.

**Weaknesses:**

1. Although the motivation is illustrated clearly, this paper does not provide good review of the literature, given the fact that the vision community has made great success for weakly-supervised segmentation tasks.  For example, I list some reference by searching through google scholar.
    * Weakly-supervised semantic segmentation:
       -  Token contrast for weakly-supervised semantic segmentation. Ru et al. CVPR 2023
       -  Ts-cam: Token semantic coupled attention map for weakly supervised object localization. Gao et al. ICCV 2021.
       -  Multi-class token transformer for weakly supervised semantic segmentation. Xu et al. CVPR 2022.
       -  Learning affinity from attention: End-to-end weakly-supervised semantic segmentation with transformers. Ru et al. CVPR 2022.
       -  Universal Weakly Supervised Segmentation by Pixel-to-Segment Contrastive Learning. ICLR 2021.
       - ScribbleSup: Scribble-Supervised Convolutional Networks for Semantic Segmentation. Lin et al. CVPR 2016.
    * Weakly-supervised instance segmentation
       -  Weakly supervised instance segmentation using class peak response. Zhou et al. CVPR 2018.
       -  Simple does it: Weakly supervised instance and semantic segmentation. Khoreva et al. CVPR 2017.
       -  Boxinst: High-performance instance segmentation with box annotations. Tian et al. CVPR 2021.

2. Generally speaking, the proposed machineries--multi-augmentation fusion and entropy-based filtering, are not novel.  These techniques are commonly used in weakly-supervised segmentation literature, e.g. AffinityNet[1].  The contribution of this paper is more like to employ SAM to tackle weakly-supervised concealed segmentation tasks.

3. I think the major contribution of this paper is not highlighted.  If the authors believe multi-feature grouping module is the major contribution, then demonstrating on the concealed object segmentation is not enough.  It would be essential to show its efficacy on other segmentation tasks.  If the authors believe the employment of SAM is the major contribution, then I think the novelty of this paper is not impressive.  The multi-augmentation fusion and entropy-based filtering tricks are well known in the literature.  It would be nice to see what are the distinctive perspectives or challenges when deploying SAM on the tasks.  Overall, the proposed method is reasonable but not novel enough.

4. The authors do not describe the limitations of the proposed method.

[1]: Learning Pixel-level Semantic Affinity with Image-level Supervision for Weakly Supervised Semantic Segmentation. Ahn et al. CVPR 2018.

**Questions:**

1.  How do you use scribbles as prompts to the SAM?  It would also be nice to include a brief formulation about how SAM generate masks for self-completeness.

2. I think the formulation in Sec 3.2 might be wrong (as far as I understand).  In my understanding, just like slot attention, the cluster prototypes $P$ aggregates pixel features with weighted-mean pixel-to-prototype attention. (See Eqn 1 and 2 in Slot Attention paper)
    * In Eqn 7, $A$ should be $A = \frac{1}{\sqrt{C}} K Q^\top$ instead
    * In Eqn 8, $D$ should be $D_{i, j} =\frac{\bar{A_{i, j}}}{\sum_l \bar{A_{l, j}}}$
    * How to broadcast cluster prototype to 2D grid?
    * What is $S$ in Eqn 10?

3. I think it would be interesting to see what are learned in the cluster prototype at different granularity.  It would be nice if we can see the visualization of feature similarity (or attention) among cluster prototype and grid features for $N_1$ and $N_2$.

**Limitations:**

The authors do not describe the limitations of the proposed method.

---

> ### Author Rebuttal · Authors · 2023-08-09
>
> Thanks for the valuable comments. Unless otherwise specified, all experiments are conducted on the *COD10K* dataset for the COD task to save space.
>
> **Q1: Literature review.**
>
> We focused on concealed object segmentation and mainly discussed and compared existing methods in this field. However, we agree that it is better to have the literature review in a broader context and includes methods for weakly-supervised segmentation. While we have cited and discussed 8 existing methods related to weakly-supervised segmentation, we will add a subsection to fully review the weakly-supervised segmentation task in the related work of the revised version.
>
> **Q2: Novelties.**
>
> WS-SAM offers the first solution to address the WSCOS task that is orthogonal to existing methods by leveraging the power of SAM. We are inspired by the impressive capability of SAM on natural scene images and extend it to conceal object images by designing multiple effective techniques to address the limitations. Extensive experiments demonstrate that our WS-SAM is a general and robust framework that can be applicable to any segmenter and promote their performance in the COS task. More concrete differences are as follows.
>
> - Although our entropy-based strategy is similar to that in AffinityNet [52], we further propose the multi-augmentation fusion and image-level selection to improve the quality of the pseudo label, whose advancements are proved in Table 3 by correspondingly bringing **4.6\%** and **4.9\%** improvements.
>
> - Unlike AffinityNet [52] using augmentation for self-ensemble, our multi-augmentation fusion aims to fully explore the potential of SAM to balance the accuracy and completeness of pseudo labeling (line 138). To achieve this, we further fuse the segmentation masks for different augmented images and acquire a more reliable pseudo label, which promotes the performance of our method.
>
> - We replace our refinement strategies counterpart with those in AffinityNet [52] and validate the effect of our WS-SAM.
>
> **Q3: Contributions.**
>
> Both WS-SAM and MFG are designed out of the characteristics of weak supervision and the COS task. They can jointly solve the weakly-supervised COS task.
>
> - WS-SAM: Given that SAM struggles in the COS task, directly using pseudo labels generated by SAM to supervise the segmenter is a sub-optimal solution. As shown in **Table A1** in the attained PDF, the segmenter trained in this manner performs poorer than that trained with the strategy of SCOD. To address this, we propose a series of refinement strategies, including multi-augmentation fusion, pixel-level weighting, and image-level selection, to fully explore the potential of SAM and thus provide reliable guidance for segmenter training, leading to improved performance. The effect of those strategies is proved in Table 3. Besides, our WS-SAM is a general framework applicable to various segmenters, whose superiority has been widely proven in **Tables A1, A5, A9**.
>
> - MFG: Given the intrinsic similarity problem in COS, we propose to address the COS task with feature grouping and introduce MFG to evacuate critical cues at different granularities, promoting feature coherence. The effect of our MFG has been proven in Table 4. We further demonstrate that our MFG is a plug-and-play module and can enhance the capability of existing segmenters in **Table A3**. Besides, as shown in **Tables A7, A8**, our segmenter equipped with the MFG is a lightweight network but achieves comparative performance with existing SOTAs in full supervision (**Tables 5, A6**).
>
> **Q4: Limitations.**
>
> We have discussed our limitations, including our WS-SAM framework and MFG, in Sec. C of the Supp. We will include this discussion in the main text in the revised version.
>
> **Q5: The use of scribble and the formulation of SAM.**
>
> - As shown in **Fig. A2**, we use a nine-box strategy to sample critical points in the scribble, which can maximize the conveyance of the scribble information to SAM, resulting in more accurate pseudo-labels (line 231).
>
> - For point supervision, the formula about how SAM generates masks is defined in Eq. 1. For scribble supervision, the corresponding formulation is defined as:
> $$\mathbf{M}_i^k=\mathrm{SAM}(\mathbf{X}^k_i,\\{\mathbf{Y}\_i^{f,c}\\}\_{c=1}^C,\\{\mathbf{Y}\_i^{b,c}\\}\_{c=1}^C),$$
> where $\\{\mathbf{Y}\_i^{f,c}\\}\_{c=1}^C$, $\\{\mathbf{Y}_i^{b,c}\\}\_{c=1}^C$ denote prompts from foreground and background that are sampled from the scribble annotation $\mathbf{Y}_i$. We will add this in the revised version.
>
> **Q6: Formulation in Sec 3.2 and implementation details.**
>
> - We double-checked the code and confirm that you are right. The transpose operator in Eq. 7 should be applied on $\\mathbf{Q}$, and the $\\mathbf{D}$ in Eq. 8 should be calculated from $\\bar{\\mathbf{A}}$. We will correct this in the revised version.
>
> - Following [31], We use the broadcast mechanism to broadcast the cluster prototype to a 2D grid, i.e., the prototype is copied horizontally and vertically several times to form a 2D grid. We will release our code.
>
> - $S$ in Eq. 10 means Sigmoid. We will clarify this in the revised version.
>
> **Q7: Visualizations.**
>
> We explore how MFG facilitates complete segmentation and multiple-object segmentation. We visualize the similarity among group prototypes and grid features in **Fig. A1**. For complete segmentation, we observe that each single group can excavate local correlation within the individual object, helping complete segmentation through the synergy of multiple groups. Besides, we observe that every group can seek global coherence across multiple objects by aggregating those similar features, thus improving the accuracy in multiple-object segmentation. **Fig. A1** shows that clustering prototypes can learn to aggregate similar features and achieve more fine-grained aggregation at a higher level of granularity. Thus, MFG can extract critical cues by fusing the multi-scale grouping features.
>
> [52] AffinityNet, CVPR, 2018.

---

> > ### Comment · Reviewer_DmES · 2023-08-12
> >
> > Thanks for the rebuttal.  I think the authors have done good job in the rebuttal.   The feedback has addressed most of my concern, except Q2 and Q3.
> >
> > The author claims multi-augmentation fusion is novel from AffinityNet[52], yet, the statement is not true.  In fact, multi-scale augmentation during inference is commonly used in segmentation tasks.  For example, in [AffinityNet's public code](https://github.com/jiwoon-ahn/psa/blob/9493e59bef16687ecb4821387e38e6460857f508/infer_cls.py#L40-L45), they do aggregate CAM from different scales.  SEAM[Ref1] imposes consistency regularizations among augmentations.  Not to mention that, DeepLab[Ref2] uses multi-scale fusion to bootstrap testing performance (see Table 4).  I'm not convinced that *multi-augmentation fusion* proposed in this paper is novel.
> >
> > On the other hand, demonstrating the efficacy of MFG only on the COS tasks seems not enough.  In my understanding, I don't think COS is special from other image segmentation tasks.  Instead, COS imagery are a special subset of common images.  As a result, it would be a plus if the authors can demonstrate that MFG also works on more general imagery.
> >
> >
> > [Ref1] Self-supervised Equivariant Attention Mechanism for Weakly Supervised Semantic Segmentation. Wang et al. CVPR 2020.
> >
> > [Ref2] DeepLab: Semantic Image Segmentation with Deep Convolutional Nets, Atrous Convolution, and Fully Connected CRFs. Chen et al.

---

> > > ### Author Response · Authors · 2023-08-15
> > > **Further response to Reviewer DmES (1/2).**
> > >
> > > **Q2: Novelties of multiple-augmentation fusion.**
> > >
> > > Our multi-augmentation fusion (MAF) strategy differs from AffinityNet [52] and DeepLab [53] in that we perform MAF in the **training phase**, while AffinityNet and DeepLab do that in the **test phase**.
> > > In addition, since our masks are generated from SAM when with scribbles as supervision, we propose to generate more masks for fusion by sampling different versions of points from scribbles. While SEAM [54] also employs augmentation in training, it, however, aims to enforce consistency between an image and its augmented views for the predictions. Instead, we employ multiple augmentations for result fusion. To further demonstrate the difference, we conducted an experiment replacing our multi-augmentation fusion with the augmentation technique utilized in SEAM [54]. We can see a noticeable performance drop after the replacement. This further evidences the advantages of our MAF strategy for the WSCOS task.
> > > |Scribble|$M\downarrow$|$F_\beta\uparrow$|$E_\phi\uparrow$|$S_\alpha\uparrow$|
> > > |-|-|-|-|-|
> > > |Ours+SEAM|0.044|0.683|0.843|0.766|
> > > |Ours+WS-SAM|0.038|0.719|0.878|0.803|
> > >
> > > We would like to stress that the differences between our method and the two methods, *i.e.*, AffinityNet [52] and DeepLab [53], are more than they look to be. Rather than directly borrowing and adapting techniques used by existing methods, the multiple-augmentation fusion strategy is particularly designed by considering the uniqueness of employing SAM to solve the WSCOS problem. We are rooted in the observations that upon feeding a sequence of images augmented from the same original source into SAM, this method yields multiple masks characterized by notable variations in shape. Notably, these divergent masks exhibit regions of overlap, as evidenced by Fig. 2 in the manuscript and Fig. 4 in the supplementary materials. Intriguingly, these overlapping areas consistently demonstrate reliable predictions by SAM, undeterred by image transformations, often aligning with accurately predicted foreground regions. Furthermore, these masks complement each other, such that some foreground regions missed by one mask can be found in other masks. This realization underscores the rationale behind our approach of fusing segmentation masks derived from diverse augmented images. Such fusion culminates in a consolidated mask that transcends the individual masks in terms of reliability due to its ensemble nature, encapsulating insights drawn from a spectrum of augmented images.
> > >
> > > In summation, our multi-augmentation fusion strategy is meaningful to the field of weakly-supervised COS. It effectively tackles the challenge of generating stable masks in concealed scenarios, setting a robust foundation to improve the segmentation performance.
> > >
> > > **Q3: Superiority of our MFG on general segmentation tasks**
> > >
> > > We prove that our MFG is a plug-and-play module in multiple segmentation tasks, including salient object detection (SOD), semantic segmentation, and concealed object segmentation (COS). In specific, we add our MFG to existing state-of-the-art segmenters in the corresponding tasks following the practice in our method and see whether or not the addition of MFG will result in performance improvement. Note that the new segmenters equipped with our MFG are still trained with their original training strategies without any modifications.
> > >
> > > - Salient object detection. We report the results in the  *DUTS-test* dataset. In the weakly-supervised setting, MFG improves existing cutting-edge segmenters by **3.9\%** (TEL) and **4.9\%** (SCOD). For those segmenters trained with full supervision, MFG reinforces their performance by **3.2\%** (PGNet [55]) and **4.0\%** (MENet [56]).
> > >
> > > |Scribble in SOD|$M\downarrow$|$F_\beta\uparrow$|$E_\phi\uparrow$|$S_\alpha\uparrow$|
> > > |-|-|-|-|-|
> > > |TEL [34]|0.045|0.842|0.901|0.863|
> > > |TEL+MFG|0.041|0.869|0.920|0.874|
> > > |SCOD [29]|0.046|0.830|0.892|0.855|
> > > |SCOD+MFG|0.041|0.862|0.918|0.872|
> > >
> > > |Full in SOD|$M\downarrow$|$F_\beta\uparrow$|$E_\phi\uparrow$|$S_\alpha\uparrow$|
> > > |-|-|-|-|-|
> > > |PGNet [55]|0.027|0.903|0.922|0.911|
> > > |PGNet+MFG|0.025|0.923|0.941|0.920|
> > > |MENet [56]|0.028|0.895|0.937|0.905|
> > > |MENet+MFG|0.025|0.919|0.949|0.917|
> > >
> > > - Weakly-supervised semantic segmentation. Following AFA [57], we conduct experiments on *PASCAL VOC 2012* and *MS COCO 2014* datasets in the weakly-supervised semantic segmentation task to verify the superiority of our MFG, where the two datasets are abbreviated as *VOC* and *COCO* in the table. Under weak supervision, MFG boosts the performance of the original segmenters by **3.1\%** (AFA [57]), **1.4\%** (AEFT [58]), **1.7\%** (ToCo [59]), and **3.8\%** (OCR [60]), where OCR is deployed on AffinityNet [52].
> > >
> > > |Weak in semantic (mIoU (\%))|*VOC\_val*|*VOC\_test*|*COCO\_val*|
> > > |-|-|-|-|
> > > |AFA [57]|66.0|66.3|38.9|
> > > |AFA+MFG|68.1|68.9|39.8|
> > > |AEFT [58]|70.9|71.7|44.8|
> > > |AEFT+MFG|72.0|72.8|45.3|
> > > |ToCo [59]|69.8|70.5|41.3|
> > > |ToCo+MFG|70.9|71.7|42.0|
> > > |OCR [60]|64.9|65.2|30.5|
> > > |OCR+MFG|67.2|68.0|31.6|

---

> > > > ### Author Response · Authors · 2023-08-15
> > > > **Further response to Reviewer DmES (2/2).**
> > > >
> > > > - Semi-supervised semantic segmentation. Following SemiCVT [61], the experiments on semi-supervised semantic segmentation are conducted on *VOC\_blender* and *Cityscapes* datasets with 1/16 labeled training data. As shown in the table, MFG generally improves the segmenters by **3.4\%** (GCT [62]), **3.7\%** (CPS [63]), **3.5\%** ($\text{U}^{\text{2}}\text{PL}$ [64]), and **2.1\%** (SemiCVT [61]). Those improvements are great in the field of semi-supervised semantic segmentation.
> > > >
> > > > |Semi in semantic (mIoU (\%))|*VOC\_blender*|*Cityscapes*|
> > > > |-|-|-|
> > > > |GCT [62]|70.9|66.8|
> > > > |GCT+MFG|72.3|69.9|
> > > > |CPS [63]|74.5|69.8|
> > > > |CPS+MFG|76.2|73.3|
> > > > |$\text{U}^{\text{2}}\text{PL}$ [64]|77.2|70.3|
> > > > |$\text{U}^{\text{2}}\text{PL}$+MFG|78.9|73.8|
> > > > |SemiCVT [61]|78.2|72.2|
> > > > |SemiCVT+MFG|79.1|74.4|
> > > >
> > > > - Fully-supervised semantic segmentation. Following TSG [65], experiments are implemented on *Pascal Context* and *ADE20K* datasets. As reported in the table, MFG improves existing cutting-edge segmenters by **2.3\%** (MaskFormer [66]), **1.6\%** (SenFormer [67]), **2.6\%** (Mask2Former [68]), and **2.4\%** (TSG [65]).
> > > >
> > > > |Full in semantic (mIoU (\%))|*Pascal Context*|*ADE20K*|
> > > > |-|-|-|
> > > > |MaskFormer[66]|53.3|46.7|
> > > > |MaskFormer+MFG|54.5|47.8|
> > > > |SenFormer[67]|53.2|46.0|
> > > > |SenFormer+MFG|54.1|46.7|
> > > > |Mask2Former[68]|54.6|47.7|
> > > > |Mask2Former+MFG|56.3|48.7|
> > > > |TSG[65]|54.5|47.5|
> > > > |TSG+MFG|55.8|48.6|
> > > >
> > > > - Concealed object segmentation. In **Table A3** in the attained PDF, MFG improves the performance of segmenters by **2.1\%** (TEL [34]) and **3.5\%** (SCOD [29]). We further evaluate the effect of our MFG in the COS task and report the results in the *COD10K* dataset of the COD task with full supervision. As shown in the table, segmenters equipped with MFG generally surpass their original versions by **3.2\%** (FGANet [15]) and **3.0\%** (FEDER [51]). Note that SegMaR (CVPR22) [17] overall surpasses LSR (CVPR21) [69] by **1.2\%**, the gains brought by our MFG are significant.
> > > >
> > > > |Full in COS|$M\downarrow$|$F_\beta\uparrow$|$E_\phi\uparrow$|$S_\alpha\uparrow$|
> > > > |-|-|-|-|-|
> > > > |FGANet[15]|0.032|0.708|0.894|0.801|
> > > > |FGANet+MFG|0.030|0.725|0.917|0.813|
> > > > |FEDER[51]|0.032|0.715|0.892|0.810|
> > > > |FEDER+MFG|0.030|0.732|0.911|0.819|
> > > >
> > > > To sum up, the proposed MFG has exhibited remarkable efficacy across a spectrum of challenging tasks, encompassing semantic segmentation, SOD, and even highly intricate COS tasks. It is noteworthy that the versatility of our MFG extends to its compatibility with diverse supervision strategies, spanning weak, semi, and full supervision scenarios, thereby substantially elevating segmentation performance. This capacity is a direct outcome of MFG's intrinsic attributes, including the exploration of foreground-background region coherence and feature grouping across multiple hierarchical levels. This distinct feature grouping approach holds applicability to a wide array of image segmentation tasks. Notably, while the segmenters' feature extraction capacity might be comparatively modest in instances like weak or semi-supervised settings, MFG adeptly capitalizes on the extracted discriminative cues. It achieves this through a multi-scale grouping mechanism that effectively fuses unattended features with the extracted ones. Consequently, segmenters fortified with MFG are aptly equipped to mitigate the issue of incomplete segmentation by augmenting local correlations within individual objects. Concurrently, they facilitate the segmentation of multiple objects by actively seeking global coherence across multiple entities. This holistic approach culminates in the strengthening of segmentation performance in multifaceted scenarios. These assertions are notably evidenced in **Fig. A1**, which graphically substantiates the advantages conferred by our MFG in enhancing both local and global segmentation outcomes.
> > > >
> > > > We will add the related experiments to our revised version.
> > > >
> > > > [53] DeepLab, TPAMI, 2017.
> > > >
> > > > [54] SEAM, CVPR, 2020.
> > > >
> > > > [55] PGNet, CVPR, 2022.
> > > >
> > > > [56] MENet, CVPR, 2023.
> > > >
> > > > [57] AFA, CVPR, 2022.
> > > >
> > > > [58] AEFT, ECCV, 2022.
> > > >
> > > > [59] ToCo, CVPR, 2023.
> > > >
> > > > [60] OCR, CVPR, 2023.
> > > >
> > > > [61] SemiCVT, CVPR, 2023.
> > > >
> > > > [62] GCT, ECCV, 2020.
> > > >
> > > > [63] CPS, CVPR, 2021.
> > > >
> > > > [64] $\text{U}^{\text{2}}\text{PL}$, CVPR, 2022.
> > > >
> > > > [65] TSG, CVPR, 2023.
> > > >
> > > > [66] MaskFormer, NeurIPS, 2021.
> > > >
> > > > [67] SenFormer, BMVC, 2022.
> > > >
> > > > [68] Mask2Former, CVPR, 2022.
> > > >
> > > > [69] LSR, CVPR, 2021.

---

> > > > > ### Comment · Reviewer_DmES · 2023-08-15
> > > > >
> > > > > Thanks for the update.  I think these comments address my concern in Q2 and Q3.
> > > > >
> > > > > The difference between AffinityNet, DeepLab, SEAM, and the proposed method is clearly illustrated in the response.  I didn't notice that SAM is a mask proposal framework: given different augmentations, the proposals will be different.  On the other hand, AffinityNet, DeepLab, and SEAM are image parsing framework.  The segmentation do not change drastically w.r.t augmentations.  It will be helpful if the authors can highlight the difference in the paper.
> > > > >
> > > > > Also, the efficacy of proposed MFG is clearly demonstrated in the comments, which not only works in WCOS tasks, but also other segmentations tasks.  It is a general and effective modules.
> > > > >
> > > > > I increase my ratings to accept based on these strong results.

---

> > > > > > ### Author Response · Authors · 2023-08-16
> > > > > > **Thanks for recognizing the value of our work!**
> > > > > >
> > > > > > We thank the reviewer for recognizing the value of our work, *i.e.*, the proposed **WS-SAM framework** and **MFG module**, to the field of COS. We appreciate the suggestion and will further highlight the differences between our method and existing weakly-supervised approaches in the revised version.

---

### Official Review · Reviewer_ka5E · 2023-07-03

**Soundness:** 3 good
**Presentation:** 3 good
**Contribution:** 3 good
**Rating:** 5
**Confidence:** 5

**Summary:**

This paper tackles the problem of Weakly-Supervised Concealed Object Segmentation (WSCOS). The first strategy is using SAM, which is a vision foundation model for generic segmentation. To handle the miss-segmentation of SAM, the authors propose techniques using multi-augmentation and entropy-based scoring. In addition, to deal with the similarity challenge of FG and BG, Multi-scale Feature Grouping (MFG) is proposed. The overall experiments show that the proposed method achieves state-of-the-art performance.

**Strengths:**

1. Using SAM is one of the promising directions in the field of segmentation. It is good to see a SAM-based weakly-supervised segmentation approach.
2. Entropy-based weighting and selection methods are technically sounding.

**Weaknesses:**

1. It seems that the baseline of the proposed method is already quite higher than the existing methods. Could the authors provide the backbone architecture of the proposed method and existing ones, and match them for fair comparison?

2. SAM also can be used to refine the pseudo label. One of the most simple approaches is using the conventional WSCOS method and then refining the obtained pseudo label with SAM. Since the proposed method is integrated with SAM in a learning phase, it should achieve better performance than the refining strategy. Please provide the comparison in detail.

3. I want to check the performance of using SAM (PLW+ILS) alone, without MFG. I'm not sure but the gain of MFG can be overlapped with that of using SAM.

4. Inference using SAM is quite heavy in terms of both time and memory. This drawback is particularly pronounced when inferring images with multiple augmentations. This computational burden should be addressed in this paper.

**Questions:**

Authors mention that directly using SAM sometimes miss-segment the confusing regions. Have the authors tried the segment-everything option of SAM? I'm not sure about it, but in such confusing cases, this option usually provides better results than using point (or scribble) prompts.

---

> ### Author Rebuttal · Authors · 2023-08-09
>
> Thanks for the valuable comments. Unless otherwise specifically stated, all experiments are conducted on the *COD10K* dataset for the  COD task to save space.
>
> Q1: Fair comparison with the same backbone.
>
> Q1-A: In Table 3, ''Baseline'' denotes using SAM to generate only one segmentation mask from one training image without augmentations for model training (line 281) rather than the backbone architecture of our segmenter. We further provide the results of TEL and SCOD with our backbone architecture, which are denoted as TEL' and SCOD'. In the table below, our segmenter still outperforms the compared methods with the same backbone.
> |Scribble|$M\downarrow$|$F_\beta\uparrow$|$E_\phi\uparrow$|$S_\alpha\uparrow$|
> |-|-|-|-|-|
> |TEL'|0.056|0.635|0.837|0.716|
> |SCOD'|0.047|0.642|0.833|0.736|
> |Ours|0.038|0.719|0.878|0.803|
>
> Q2: Refine pseudo labels with SAM.
>
> Q2-A: We conduct detailed experiments to explore the capacity of SAM in the refinement of pseudo labels.
>
> - We employ a pretrained segmenter, e.g., SCOD [29] or our segmenter, to generate coarse pseudo labels and treat those pseudo labels as dense prompts to SAM for further refinement. We then use the refined pseudo labels to directly supervise our segmenter without the usage of our refinement strategy in the training phase. Note that the modifications only lie in that we add a pretrained segmenter to provide dense prompts for SAM and remove the proposed refinement strategy for the generated pseudo labels.
>
> - In the implementation, we provide two compared methods, termed Ours-1 and Ours-2. Ours-1 and Ours-2 mean using pretrained SCOD and our segmenter to generate the initial pseudo labels, respectively. As shown in **Table A9** in the attached file, we report the results in both scribble and point supervision and find that our segmenter achieves the best results when being trained with our WS-SAM framework.
>
> - Having analyzed the results, we discover that the results of Ours-1 and Ours-2 both surpass those of the ''Baseline'' reported in Table 3, which indicates that refining pseudo labels with SAM can really promote the quality of pseudo labels. However, in **Table A9**, when further reinforced by our WS-SAM framework, the leaderships of Ours-1 and Ours-2 are mitigated and have a comparative performance with our segmenter. This indicates the effect of our WS-SAM in generating high-quality pseudo labels to supervise the segmenter.
>
> |Scribble|$M\downarrow$|$F_\beta\uparrow$|$E_\phi\uparrow$|$S_\alpha\uparrow$|
> |-|-|-|-|-|
> |Ours-1|0.047|0.688|0.851|0.765|
> |Ours-2|0.046|0.693|0.855|0.772|
> |Ours-1+WS-SAM|0.038|0.716|0.877|0.805|
> |Ours-2+WS-SAM|0.038|0.720|0.873|0.802|
> |Ours+WS-SAM|0.038|0.719|0.878|0.803|
>
> |Point|$M\downarrow$|$F_\beta\uparrow$|$E_\phi\uparrow$|$S_\alpha\uparrow$|
> |-|-|-|-|-|
> |Ours-1|0.053|0.630|0.811|0.732|
> |Ours-2|0.050|0.641|0.820|0.747|
> |Ours-1+WS-SAM|0.039|0.700|0.853|0.782|
> |Ours-2+WS-SAM|0.039|0.695|0.855|0.790|
> |Ours+WS-SAM|0.039|0.698|0.856|0.790|
>
> Q3: Effect of MFG.
>
> Q3-A: MAF in Table 3 denotes multi-augmentation fusion rather than MFG. Table 4 indicates that the use of MFG brings a **6.0\%** improvement when using the proposed MS-SAM framework. In the following table (**Table A3**), we further add our MFG to other segmenters and find performance gains.
>
> |Scribble|$M\downarrow$|$F_\beta\uparrow$|$E_\phi\uparrow$|$S_\alpha\uparrow$|
> |-|-|-|-|-|
> |TEL[34]|0.057|0.633|0.826|0.724|
> |TEL+MFG|0.054|0.642|0.833|0.731|
> |SCOD [29]|0.049|0.637|0.832|0.733|
> |SCOD+MFG|0.046|0.672|0.840|0.744|
>
> Q4: Computational burden from SAM.
>
> Q4-A: SAM and the multi-augmentation strategy are only employed in the training phase. Additionally, we propose a scheme that generates all augmented images offline using SAM and then fuses the final supervised results with randomly selected images from these sets during online training. This is the same as using multiple augmentations for online inference with SAM when we are training the segmenter. Therefore, such behavior brings no performance decrease but greatly reduces the computational burden required. The computational burden of the proposed pseudo-label refinement strategy, including pixel-level weighting and image-level selection, is almost negligible, thus ensuring our efficiency. In **Table A8**, we further provide our parameters and FLOPs in such a scheme and find that we achieve the best performance among all compared methods in the setting of weak supervision.
>
> |Weak|WSSA[32]|SCWS[33]|TEL[34]|SCOD[29]|Ours|
> |-|-|-|-|-|-|
> |Parameters|53.31|58.83|48.62|49.93|31.12|
> |FLOPs|65.87|77.18|69.12|58.63|40.06|
>
> Q5: Can the segment-everything option of SAM work in COS?
>
> Q5-A: When using the segment-everything (SE) option, SAM outputs a series of binarized segmentation results and sorts them in order of IoU scores. We choose the segmentation mask with the highest IoU score and refine the mask with our WS-SAM framework. From the table (**Table A10**), our segmenter achieves better results in both scribble and point supervision. This is because point or scribble actually provides important localization information. Whereas, the output of SE with the highest IoU score may not have accurate localization, leading to the generation of low-quality pseudo-labels.
>
> |Scribble|$M\downarrow$|$F_\beta\uparrow$|$E_\phi\uparrow$|$S_\alpha\uparrow$|
> |-|-|-|-|-|
> |SE|0.044|0.662|0.849|0.762|
> |Ours|0.038|0.719|0.878|0.803|
>
> |Point|$M\downarrow$|$F_\beta\uparrow$|$E_\phi\uparrow$|$S_\alpha\uparrow$|
> |-|-|-|-|-|
> |SE|0.046|0.646|0.820|0.749|
> |Ours|0.039|0.698|0.856|0.790|

---

> > ### Comment · Reviewer_ka5E · 2023-08-17
> >
> > Thanks for the rebuttal.
> > All of my concerns are appropriately addressed.
> > Since the overall impact of this paper is not that huge, I'll remain my rating as borderline accept.
> > But I still think this paper is good.

---

> > > ### Author Response · Authors · 2023-08-17
> > > **Thanks for recognizing the value of our work!**
> > >
> > > We express our sincere appreciation to the reviewer for acknowledging the substantive impact of our contribution, notably the **WS-SAM framework** and the **MFG module**, within the domain of COS and for maintaining a positive rating.
> > >
> > > WS-SAM offers the **first solution** to address the WSCOS task that is **orthogonal to existing methods** by leveraging the power of SAM. Moreover, our proposed WS-SAM framework and the plug-and-play MFG module can **significantly enhance the performance of existing state-of-the-art segmenters**. Notably, MFG holds applicability to a wide array of image segmentation tasks, including **concealed object segmentation**, **salient object detection**, and **semantic segmentation**. (See ''**Further response to Reviewer DmES**'' for more details.)

---

### Official Review · Reviewer_yhjR · 2023-07-06

**Soundness:** 2 fair
**Presentation:** 1 poor
**Contribution:** 2 fair
**Rating:** 5
**Confidence:** 4

**Summary:**

This paper investigates Concealed Object Segmentation based on the recently proposed foundation model SAM, which provides dense annotation for learning. To alleviate the la-bel noise caused by imperfect segmentation results of SAM on concealed objects, they adopt pixel-level and image-level uncertainty-based strategies to weight predictions and select reasonable training samples. To tackle the intrinsic similarity problem in the task, they design a multi-scale grouping module to group features from different levels. Further, they conduct experiments on commonly used datasets, which demonstrates better performance.

**Strengths:**

+ To alleviate the sparse-annotated challenge, this work utilizes the recently proposed foundation model SAM to augment the label, along with entropy-based acquisition strategies, to alleviate the impact of low-quality segmentation masks produced by SAM.

+ To tackle the inherent similarity problem in COD/COS, they introduce multi-scale features grouping technique to group features with different granularities.

+ To demonstrate the efficacy of the proposed method, they conduct experiments on several concealed datasets compared with recent methods and ablate their design components.

+ The proposed model achieves high performance from the experimental results.

**Weaknesses:**

- ”Weakly-supervised” claim is not appropriate in this work. Since the segmentation mask generated by SAM is close to the full supervision setup but carries noise, instead of sparse as the weak-supervised task definition, such a claim here is inappropriate.

- Lack of novelty. The proposed entropy-based acquisition strategies are classic methods used in active learning to sample informative samples and semi-supervised learning tasks to generate pseudo labels. Moreover, multi-augmentation fusion is also a popular technique in semi-supervised learning works. The aforementioned existing methods may refer to the following literature (but not limited to):

a. Settles, Burr. ”Active learning literature survey.” (2009).
b. Ren, Pengzhen, et al. ”A survey of deep active learning.” ACM computing surveys (CSUR) 54.9 (2021): 1-40.
c. Iscen, Ahmet, et al. ”Label propagation for deep semi-supervised learning.” Proceedings of the IEEE/CVF conference on computer vision and pattern recognition. 2019.
d. Berthelot, David, et al. ”Mixmatch: A holistic approach to semi-supervised learning.” Advances in neural information processing systems 32 (2019).

- Literature review is not sufficient. There are more works related to the Concealed Object Detection/Segmentation task, which are not included and discussed the difference with the proposed method in the paper.

- The mask generation method in Eq.5 is wrong. First, $\tilde{E_i}$ is an entropy mask. Its value ranges from [0, ln2] rather than [0,1]. In Eq.5, the author uses $(1-\tilde{E_i})$ to get the reverse mask is not the correct way. Second, the author regards $(1-\tilde{E_i})$ as a certainty for pixels. They should use $(1-\tilde{E_i})$ as weighting coefficients and multiply $(1-\tilde{E_i})$ to cross-entropy loss to calculate the weighted loss. However, the author directly multiplies certainty $(1-\tilde{E_i})$ to label $\tilde{M_i}$, which causes the pseudo mask to no longer be 0 or 1, but a decimal point. For example, on pixel in $\tilde{M_i}$ equals 1, but it will become 0.5 when $(1-\tilde{E_i})=0.5$. Using the label=0.5 to supervise the segmentation model does not make sense. Third, the author proposes the image selection value \hat{\alpha} to remove images with high uncertainty. However, they multiply the $\hat{\alpha}$ to the pseudo mask in Eq.5, which confuses the reviewer. Because this operation will generate an all-zero mask as the pseudo mask. Obviously, using the all-zero mask to supervise modeling training brings the wrong guidance. The authors need to repolish the theoretical part of the paper.

- The experiment is not convincing. In line 241, the authors set two baselines SAM-S and SAM-P but only fine-tune the decoder part and use the automatic prompt generation strategy. This way can not show the generalization ability of SAM. The authors should adopt the SAM encoder as the backbone network, finetune the whole model and abandon the automatic prompt generation strategy. Only in this way can we understand the generalization of SAM to the COD task. Furthermore, by comparing the way of pseudo label generation and the way of encoder finetuning, the reviewer can determine whether we need better pseudo labels for COD or just a SAM backbone.

- The writing is not clear enough, especially the use of symbols is confusing. For example, $\phi_p$ in the caption of Figure 3 is not displayed in the figure. "S" in line 204 is given without any explanation.

**Questions:**

1. In the Supp., the authors describe that they adopt the encoder-decoder architecture utilized in a more recent work FEDER. But they do not compare the results with this model in their experiments.

2. The ablation in Table 4 does not show the superiority of their sophisticated design choices, where the margins among experiment results are narrow.

**Limitations:**

Please refer to the weakness part.

---

> ### Author Rebuttal · Authors · 2023-08-09
>
> Thanks for the valuable comments. Unless otherwise specified, all experiments are conducted on the *COD10K* dataset for the COD task to save space.
>
> **Q1: ''Weakly-supervised'' claim is not appropriate.**
>
> We follow SCOD [29] and use the term ''weakly-supervised'' to indicate that training data only provide weak supervision in the forms of sparse points or scribble. While it is true that we use dense masks to train our segmenter, the dense masks are generated from the sparse annotations and they are not reliable. In **Table A1** in the attached file, we show that training with these unreliable dense masks brings negative effects and only refined with our proposed techniques, the dense masks can bring performance gains.
>
> **Q2: Novelties.**
>
> WS-SAM offers the **first solution** to address the WSCOS task that is **orthogonal to existing methods** by leveraging the power of SAM. We are inspired by the impressive capability of SAM on natural scene images and extend it to conceal object images by designing multiple effective techniques to address the limitations. While these techniques may be rooted in some existing techniques used in various tasks, we are first to explore and adapt them to address our target problems. More concrete differences are as follows.
>
> - *Entropy-based acquisition strategy*: Our difference compared with semi-supervised learning methods is that we use the entropy-based metric to perform both pixel-level weighting and image-level selection, previous methods usually include either of them.
>
> - *Multi-augmentation fusion strategy*:  Different from Mixmatch [50] using self-ensemble strategies to minimize the variance among pseudo labels, our purpose is to fully explore the potential of SAM to balance the accuracy and completeness of pseudo labeling (line 138). To achieve this, we employ flipping, rotation, and scaling for image augmentations rather than the elastic deformation and salt-and-pepper noise used in Mixmatch. To verify this, we replaced our augmentation strategies with that used in Mixmatch and found a decrease in performance (**Table A2** in the attached file).
>
> **Q3: Discussion with more related works.**
>
> We have cited (~20) most recent COS methods. We will search and add more relevant ones in the revised version.
>
> **Q4: Some formulas.**
>
> We double-checked our implementation and confirmed there are some subtle differences between the implementation and the writing.
>
> - *$\tilde{\mathbf{E}}_i$ in Eq. (3)*. We calculate our entropy mask using the logarithm of base 2 instead of e. Therefore, the value ranges of $\tilde{\mathbf{E}}_i$ is [0,1]. We will highlight this in the revised version.
>
> - *Eqs. (5) and (11)*. The idea is to use a mask to assign higher weights to reliable pixels and discard low-quality pseudo labels when calculating the cross-entropy loss. We implement this as the weighted cross-entropy loss. However, in our writing, we thought that this practice was equivalent to multiplying the mask with the binarized pseudo labels. However, after carefully checking the math, we realize that there is a subtle difference between these two formulations. The weighted cross-entropy loss applies the mask outside of the logarithmic operator, but multiplying the mask with the binarized pseudo labels applies the mask inside the logarithmic operator, which causes problems as you described. So, the formulations for Eq (5) and Eq. (11) that match our implementation should be
>
> $$\hat{\mathbf{Y}}_i=(1-\tilde{\mathbf{E}}_i)\times\hat{\mathbf{M}}_i.$$
>
> $$L=\frac{1}{N_s}\sum_{(\mathbf{X}_i,\mathbf{Y}_i)\sim\mathcal{S}}L\_{pce}(\mathbf{Y}_i',\mathbf{Y}_i)+{\hat{\mathbf{Y}}_i}L\_{ce}(\mathbf{Y}'_i, \tilde{\mathbf{M}}_i)+{\hat{\mathbf{Y}}_i}L\_{IoU}(\mathbf{Y}'_i, \tilde{\mathbf{M}}_i).$$
>
> We will correct the two equations in the revised version and release our code.
>
> **Q5: Another learnable SAM baseline.**
>
> We construct the baseline, SAM-L, as described and train SAM-L for the COD, PIS, and TOD tasks. We report the results from *CVC-ColonDB* in PIS and from *GDD* in TOD. **Table A4** (in the attached file) shows that SAM-L still struggles in the COS task. This is mainly because of the sparse annotation and the limited training data, which leads to overfitting when training SAM. Such discovery indicates that directly utilizing SAM to address those extreme tasks, such as COS, maybe a sub-optimal solution.
>
> Similar to Table 5 in the Supp, we explore the quality of pseudo labels generated by SAM-L, report the results in **Table A5**, and achieve a similar conclusion to that in lines 115-122 in the Supp. Such findings verify that our WS-SAM framework is crucial to SAM in addressing the weakly-supervised COS task.
>
> **Q6: Writing clarification.**
>
> $\Phi_P$ is a generic term that stands for $\Phi_{N_1}$ and $\Phi_{N_2}$. $S$ in line 204 denotes Sigmoid. We will modify them accordingly.
>
> **Q7: Comparison with FEDER.**
>
> We compare our segmenter with FEDER [51] in weak supervision (with WS-SAM) and full supervision. **Table A6** in the attached PDF shows that our method outperforms FEDER in weak supervision and has comparable performance to FEDER in full supervision. Besides, our method is 28.3\% lighter than FEDER overall in the full supervision (see **Table A7**).
>
> **Q8: Narrow gain of MFG.**
>
> Table 4 shows that MFG overall brings an improvement of **6.0\%**, which is significant in this field.  The absolute value differences look small because the used metrics tend to scale down the performance difference.  Furthermore, employing feature grouping, multiscale strategy, and weighted gate mechanism can correspondingly improve the performance by **0.9\%**, **3.1\%**, and **1.5\%**, which are recognized as significant for the used metrics in the field. In **Table A3**, we further integrate our MFG with other segmenters and get performance gains.
>
> [48] A survey of deep active learning, CSUR, 2021.
>
> [49] LPSSL, CVPR, 2019.
>
> [50] Mixmatch, NIPS, 2019.
>
> [51] FEDER, CVPR, 2023.

---

> > ### Comment · Reviewer_yhjR · 2023-08-19
> >
> > Thanks the author for their detailed rebuttal, particularly the experiments for the SAM model with full finetuning, which addresses my major concern. Based on these improvements, I am inclined to upgrade my rating to "borderline accept." The author's thorough response and the subsequent enhancements have strengthened the paper's quality and merit.

---

> > > ### Author Response · Authors · 2023-08-19
> > > **Thanks for recognizing the value of our work!**
> > >
> > > We wish to express our sincere appreciation to the reviewer for recognizing the substantial significance of our contribution, specifically the **WS-SAM framework** and the **MFG module**, within the realm of concealed object segmentation (COS). Your acknowledgment carries considerable weight and serves as a meaningful validation of our relentless efforts to advance this pivotal area of research.

---

> ### Comment · Area_Chair_V7za · 2023-08-18
>
> Dear Reviewer yhjR:  Could you please check if authors' extensive rebuttal has addressed your concerns?  The author-reviewer discussion period is closing soon.  Thank you!  AC

---

### Official Review · Reviewer_76do · 2023-07-08

**Soundness:** 2 fair
**Presentation:** 3 good
**Contribution:** 2 fair
**Rating:** 6
**Confidence:** 5

**Summary:**

The submission presents a method for Weakly-Supervised Concealed Object Segmentation (WSCOS) to address the challenges of segmenting objects that are well blended with their surroundings using sparsely-annotated data.
+ The method introduces a multi-scale feature grouping module that groups similar features at different scales, promoting segmentation coherence and producing accurate results for single and multiple-object images.
+ The authors leverage the Segment Anything Model (SAM) and utilize sparse annotations as prompts to generate segmentation masks for training the model.
+ Additionally, the paper proposes strategies such as multi-augmentation result ensemble, entropy-based pixel-level weighting, and entropy-based image-level selection to mitigate the impact of low-quality segmentation masks and provide more reliable supervision during training.
+ Experimental results demonstrate the effectiveness of the proposed WSCOS method.

**Strengths:**

This paper exploits the hot SAM for weakly-supervised COS, which is a good point. This paper is well-written and easy to read and understand. They verify the method on several interesting applications. Still, there have several concerns during my reviewing, please see the below sector.

**Weaknesses:**

The performance improvements may come from the strong capability and prediction of SAM, but not from the proposed weakly-supervised strategy. The observed performance of SCOD+ does not significantly surpass that of SCOD when integrated with the proposed WS-SAM framework. Thus, the proposed modules do not seem to offer a promising and significant solution to this challenging task.

The comparison might not be fair due to the introduction of an additional model (SAM) to generate weak labels. Could the authors elaborate on this, particularly in comparison to the other methods displayed in Table 1?

Several of the latest and SOTA weakly-supervised semantic segmentation methods, such as [ref1] "Weakly Supervised Semantic Segmentation via Adversarial Learning of Classifier and Reconstructor", [ref2] "Out-of-Candidate Rectification for Weakly Supervised Semantic Segmentation", and [ref3] "Weakly Supervised Semantic Segmentation via Adversarial Learning of Classifier and Reconstructor" could be applied to this challenging task.

Overall, this submission falls short of the standard for a top-tier conference due to its weak experimental validations. I am inclined to recommend borderline rejection in the first round.

**Questions:**

See above concerns.

**Limitations:**

No significant limitations.

---

> ### Author Rebuttal · Authors · 2023-08-09
>
> Thanks for the valuable comments. If not specifically stated, all experiments are conducted on the COD task with *COD10K* for space limitation.
>
> **Q1: Performance gains are from SAM, not from the weakly-supervised strategy.**
>
> While SAM has shown impressive performance for natural scene images, its performance on concealed images is far from satisfactory, even using points/scribbles as the prompts. The table presented below (**Table A1** in the attained PDF) shows that naively employing the SAM-generated masks (without using our weakly-supervised strategy) to train SCOD results in performance (SCOD+SAM) that is even lower than SCOD. This indicates that low-quality masks generated by SAM bring negative effects to train the model. On the other hand, training SCOD with our weakly-supervised strategy backed SAM, the result (SCOD+WS-SAM) is higher than SCOD. Similar observations can be found in applying SAM and WS-SAM on TEL and our segmenter. This indicates our weakly-supervised strategy successfully increases the quality of the generated masks, and enhance improve the performance.
>
> SCOD+  (SCOD+WS-SAM)  overall surpasses SCOD by **1.5\%** (scribble) and **2.4\%** (point). For some tasks, the improvement gains can reach up to **3.6\%**. While it might be true that the numeric difference looks not significant, this is partially due to the used metrics tending to scale down the performance difference. Also COD is a very challenging task such that a ~1% gain is usually recognized as a valuable gain by the community.
>
> |Scribble|$M\downarrow$|$F_\beta\uparrow$|$E_\phi\uparrow$|$S_\alpha\uparrow$|
> |-|-|-|-|-|
> |TEL[34]|0.057|0.633|0.826|0.724|
> |TEL+SAM|0.059|0.614|0.810|0.716|
> |TEL+WS-SAM|0.053|0.646|0.838|0.730|
> |SCOD[29]|0.049|0.637|0.832|0.733|
> |SCOD+SAM|0.050|0.623|0.813|0.725|
> |SCOD+WS-SAM|0.047|0.650|0.845|0.742|
> |Ours+SCOD|0.045|0.692|0.853|0.759|
> |Ours+SAM|0.052|0.674|0.838|0.737|
> |Ours+WS-SAM|0.038|0.719|0.878|0.803|
>
> **Q2: Comparison is not fair due to the use of SAM.**
>
> The proposed WS-SAM is a general framework that can be integrated with various existing methods and improve performance. WS-SAM offers a solution orthogonal to existing methods by leveraging the power of SAM. To make fair comparisons, we not only compared with the existing WSCOS method, e.g., SCOD, and also compare with the variants that integrate SAM with SCOD (SCOD+SAM) and our WS-SAM with SCOD (SCOD+WS-SAM). The results in the table above show that our proposed WS-SAM framework can improve these methods.
>
> **Q3: Comparison with existing weakly-supervised semantic segmentation methods.**
>
> We focused on concealed object detection and only compared existing methods in this field. As per your request, we train these Weakly-Supervised Semantic Segmentation (WSSS) methods (for which code is open-sourced) and get the table below (**Table A1** in the attained PDF).  We can see that these WSSS methods struggle on the challenging COD task, but their performance can be improved when empowered by our WS-SAM framework. This further substantiates the effectiveness of WS-SAM.
>
> |Scribble|$M\downarrow$|$F_\beta\uparrow$|$E_\phi\uparrow$|$S_\alpha\uparrow$|
> |-|-|-|-|-|
> |ACR[46]|0.050|0.622|0.828|0.728|
> |ACR+WS-SAM|0.048|0.637|0.836|0.735|
> |OCR[47]|0.052|0.607|0.796|0.704|
> |OCR+WS-SAM|0.048|0.636|0.830|0.727|
>
> **Q4: Weak experimental validations.**
>
> We have validated the superiority of our method on the COS task and the SOD task. The effectiveness of WS-SAM and MFG has also been demonstrated by abundant ablation studies.
>
> During the rebuttal period, we conducted comprehensive comparisons between our segmenter and several state-of-the-art (SOTA) segmenters, as highlighted in **Tables A1, A6, A7, A8**. These comparisons clearly underscore the superior performance and efficiency of our segmentor. Furthermore, we extended our assessment to verify the generalizability of our WS-SAM framework, as shown in **Tables A1, A5, A9**, effectively demonstrating its robustness. Notably, we showcased the versatility of our Module Fusion Gateway (MFG) as a plug-and-play component that significantly enhances the segmentation capabilities of existing segmenters, as indicated in **Table A3**.
>
> In our pursuit of evaluating the efficacy of WS-SAM, we engaged in comparisons with existing strategies, unequivocally establishing our framework's superiority in addressing the COS task, as evidenced in **Table A2**. To comprehensively explore the potential of SAM, we introduced multiple modifications, including a learnable SAM baseline (**Tables A4, A5**), SAM-assisted refinement of pseudo labels (**Tables A9**), and leveraging SAM's "segment-everything" opinion for concealed object segmentation (**Tables A10**). Through these meticulous explorations, we uncovered that even with these modifications, segmenters still encounter challenges in the COS task, yet our WS-SAM framework consistently elevates their performance to a leading position.
>
> Collectively, these extensive investigations decisively illustrate that WS-SAM stands as a potent and versatile framework with remarkable generalizability for the concealed object segmentation task. To enhance clarity, we have included a diagram illustrating the nine-box strategy in **Fig. A2**, along with a visualization depicting the similarity among group prototypes and grid features in **Fig. A1**. We intend to incorporate all these experimental validations into our revised version for a more comprehensive representation of our work.
>
> [45]  LSR, CVPR, 2021.
>
> [46] ACR, CVPR, 2023.
>
> [47] OCR, CVPR, 2023.

---

> > ### Comment · Reviewer_76do · 2023-08-16
> >
> > After reviewing all the materials and discussions on this page, I believe the authors have made significant efforts to address my main concerns. I am now happy to update my rating to acceptance. Best of luck to the authors!

---

> > > ### Author Response · Authors · 2023-08-16
> > > **Thanks for recognizing the value of our work!**
> > >
> > > We extend our gratitude to the reviewer for acknowledging the significance of our contribution, namely the **WS-SAM framework** and the **MFG module**, within the domain of concealed object segmentation (COS). Your recognition is deeply appreciated and serves as validation of our efforts in advancing this critical area of research.

---

### Author Rebuttal · Authors · 2023-08-09

We extend our sincere gratitude to all the reviewers (**R1**-76do, **R2**-yhjR, **R3**-ka5E, and **R4**-DmES) for their insightful and considerate reviews, which help us to emphasize the contributions of our approach. We are encouraged to hear that the reviewers found the work is well-motivated with good presentation (**R1**, **R3**, **R4**), as well as the comprehensive experimental evaluation and commendable performance (**R1**, **R2**, **R3**, **R4**).

We are delighted to see reviewers confirm our contributions to the field of concealed object segmentation. These encompass our novel weakly-supervised framework, WS-SAM, and the ingenious plug-and-play feature grouping module, MFG.

In direct response to your thoughtful comments, we have methodically addressed each point in our individual responses, and we provide a summary here:

- We compared our method with more SOTAs to underscore our superiority in performance and efficiency.

- We added experiments to verify the generalizability of our WS-SAM framework and the advancement of our plug-and-play MFG module.

- We introduced modifications to SAM to explore its potential and validate the efficacy of WS-SAM.

- We added diagrams and revised certain formulations to enhance clarity.

Thanks again for all of your valuable suggestions. We will update the paper accordingly and release our code for community study. We appreciate the reviewers' time to check our response and **hope to further discuss with the reviewers whether the concerns have been addressed or not**. If the reviewers still have any unclear parts about our work, please let us know.

---

### Decision · Program_Chairs · 2023-09-21

**Decision:**

Accept (poster)

**Comment:**

The paper addresses Weakly-Supervised Concealed Object Segmentation by utilizing SAM mask predictions over augmentations and multi-scale feature grouping.  Both technical aspects have been thoroughly validated to be helpful in diverse settings upon reviewers' questioning and authors' extensive rebuttals.  All reviewers reach the consensus of accept.